# Manipulating the oxygen reduction reaction pathway on Pt-coordinated motifs

Jiajun Zhao[1,2], Cehuang Fu [1], Ke Ye[1,2], Zheng Liang [3], Fangling Jiang[4], Shuiyun Shen[1], Xiaoran Zhao[5], Lu Ma[6], Zulipiya Shadike[1], Xiaoming Wang[7], Junliang Zhang [1✉] & Kun Jiang [1,2✉]

Electrochemical oxygen reduction could proceed via either 4e$^-$-pathway toward maximum chemical-to-electric energy conversion or 2e$^-$-pathway toward onsite $H_2O_2$ production. Bulk Pt catalysts are known as the best monometallic materials catalyzing $O_2$-to-$H_2O$ conversion, however, controversies on the reduction product selectivity are noted for atomic dispersed Pt catalysts. Here, we prepare a series of carbon supported Pt single atom catalyst with varied neighboring dopants and Pt site densities to investigate the local coordination environment effect on branching oxygen reduction pathway. Manipulation of 2e$^-$ or 4e$^-$ reduction pathways is demonstrated through modification of the Pt coordination environment from Pt-C to Pt-N-C and Pt-S-C, giving rise to a controlled $H_2O_2$ selectivity from 23.3% to 81.4% and a turnover frequency ratio of $H_2O_2/H_2O$ from 0.30 to 2.67 at 0.4 V versus reversible hydrogen electrode. Energetic analysis suggests both 2e$^-$ and 4e$^-$ pathways share a common intermediate of *OOH, Pt-C motif favors its dissociative reduction while Pt-S and Pt-N motifs prefer its direct protonation into $H_2O_2$. By taking the Pt-N-C catalyst as a stereotype, we further demonstrate that the maximum $H_2O_2$ selectivity can be manipulated from 70 to 20% with increasing Pt site density, providing hints for regulating the stepwise oxygen reduction in different application scenarios.

[1] Institute of Fuel Cells, School of Mechanical Engineering, Shanghai Jiao Tong University, Shanghai 200240, China. [2] Interdisciplinary Research Center, School of Mechanical Engineering, Shanghai Jiao Tong University, Shanghai 200240, China. [3] Laboratory of Energy Chemical Engineering, Frontiers Science Center for Transformative Molecules, School of Chemistry and Chemical Engineering, Shanghai Jiao Tong University, Shanghai 200240, China. [4] State Key Laboratory of High-Performance Ceramics and Superfine Microstructure, Shanghai Institute of Ceramics, Chinese Academy of Sciences, Shanghai 201899, China. [5] Shanghai Key Laboratory of Advanced High-Temperature Materials and Precision Forming, State Key Laboratory of Metal Matrix Composites, School of Materials Science and Engineering, Shanghai Jiao Tong University, Shanghai 200240, China. [6] National Synchrotron Light Source II, Brookhaven National Laboratory, Upton, MA NY11973, USA. [7] Department of Chemistry and Key Laboratory for Preparation and Application of Ordered Structural Materials of Guangdong Province, Shantou University, Shantou 515063, China. ✉email: junliang.zhang@sjtu.edu.cn; kunjiang@sjtu.edu.cn

Electrocatalytic oxygen reduction reaction (ORR) is an important reaction in the process of renewable energy conversion and utilization. Molecular $O_2$ can be reduced via a 4e⁻-pathway into $H_2O$ or via a 2e⁻-pathway into $H_2O_2$. The former serves as the vital reaction in proton exchange membrane fuel cells (PEMFCs) and metal-air batteries to maximize chemical energy conversion efficiency[1–3], the latter represents an environmentally benign method for the onsite production of hydrogen peroxide commodity[4–7]. Therefore, a facile ORR reaction pathway tuning is highly demanded for both fundamental mechanistic investigations and different application scenarios.

$$*O_2 + (H^+ + e^-) \rightarrow *OOH \tag{1}$$

$$*OOH + (H^+ + e^-) \rightarrow *O + H_2O \tag{2.1}$$

$$*OOH + (H^+ + e^-) \rightarrow H_2O_2 \tag{2.2}$$

Earlier theoretical studies suggest the above two ORR pathways share a common intermediate of *OOH, for which its binding strength determines the reaction products[8]. A strong binding of *OOH with a parallel orientation to electrode surface tends to dissociate O–O bond, leading to the total reduction product of $H_2O$[9]. In contrast, weak *OOH interaction with a perpendicular orientation to the surface tends to preserve O–O bond and gives rise to the production of $H_2O_2$[10,11].

Among various transition metal based ORR catalysts, Pt has the highest intrinsic activity for reducing $O_2$ into $H_2O$ toward a full chemical-to-electrical energy conversion and thus been intensively investigated as a model system for decades[12,13]. For the three low-index Pt(hkl) surfaces, the 4e⁻-pathway selectivity increases in the order of Pt(111) < Pt(100) < Pt(110) within 0.1 M $HClO_4$ at large overpotential regime[14]. For nanosized Pt, it is noted that the 2e⁻-pathway selectivity increases with decreasing Pt particle size as well as increasing particle inter-spacing[15–17]. Moreover, isolating of continuous Pt surface sites by carbon layers[18], calix[4]arene molecules[19], halogen[20,21] or cyanide[22] anions, or by alloying with a secondary metal[23–27] has been demonstrated to significantly alter the ORR pathway toward $H_2O_2$ generation. Siahrostami et al. have computationally screened a wide range of bulk alloys containing a single active element toward ORR surrounded by an inert element of Hg and Au[4]. The active element like Pt is capable of adsorbing molecular $O_2$ and reducing it to *OOH but is unable to dissociate the O−O bond due to the neighboring environment. This prediction has been experimentally verified on Pt-Hg/C to deliver a $H_2O_2$ selectivity over 90% at the potential ranging from 0.3 to 0.5 V vs. reversible hydrogen electrode (RHE)[4].

Along this line, it is very interesting to see if ultimately isolated Pt single atom sites could exhibit a sole selectivity toward 2e⁻ ORR pathway. Indeed, atomic dispersed Pt sites over TiN[28], TiC[29] and carbon nanotube (CNT)[30] substrates are reported with preferential $H_2O_2$ selectivity over 65% in acidic electrolyte and an onset potential up to 0.45 V vs. RHE. Other Pt single atom catalysts (SACs) supported on highly sulfur-doped zeolite template carbon substrate[31], AuCu metallic aerogels[32] or $CuS_x$ hollow nanosphere[33] even deliver a $H_2O_2$ selectivity above 90% over a wide potential range. Nevertheless, there're recent studies showing that Pt SACs efficiently convert $O_2$ into $H_2O$ via the 4e⁻ pathway comparable to bulk Pt but at a much lower Pt usage[34–36]. Sun et al. demonstrated a maximum single-cell power density of 0.68 W cm⁻² using carbon black supported Pt single atoms as the cathode catalyst in PEMFC, corresponding to a Pt utilization efficiency of 0.13 $g_{Pt}$ kW⁻¹[37]. This single-cell performance has been further boosted to 0.09 $g_{Pt}$ kW⁻¹ on carbon-defect-anchored Pt SAC in a latest report[38]. These conflicting observations suggest that the ORR pathway and product

selectivity on isolated Pt sites may be tailored by different reactivity of Pt central atom as arisen from different coordination environment[39–43]. How to address the above controversy, to correlate the apparent ORR performance with Pt-coordinated motifs at atomic level is therefore highly demanded.

Herein, we aim to address the above controversy with the merits of investigating Pt local coordination environment effect on the ORR products selectivity. By dispersing isolated Pt atoms over carbon nanotube substrates with different metalloid dopant, a series of Pt-X-C (X = S, C, N) motifs have been prepared, and the selectivity of 2e⁻ ORR pathway increases in the order of Pt-C < Pt-N < Pt-S moieties in acidic media. Then Pt-N-CNT is deployed as a prototype to screen the Pt sites density effect on ORR pathway tuning, the selectivity of 2e⁻ pathway decreases from 70 to 20% with increasing isolated Pt sites density from 0.7 to 11.2 µg cm⁻², providing hints for regulating the stepwise oxygen reduction in different application scenarios.

## Results

### Effect of local coordination environment on ORR selectivity.
To prepare the atomic dispersed Pt-X-CNT catalysts, a small amount of Pt cations (~ 0.03 at%) were firstly dispersed with aqueous CNT suspension, followed by lyophilization and thermal annealing with certain dopant precursor (see Experimental). Given a constant Pt loading of ~ 0.7 wt%, no metal clusters nor nanoparticles were observed from bright field transmission electron microscopy (TEM) images (Supplementary Fig. 1). High-resolution XRD patterns in Supplementary Fig. 2 also reveal the absence of any long-range ordered Pt crystalline structure or nanoparticles. Figure 1a-c show the aberration-corrected high angle annular dark field scanning TEM (HAADF-STEM) characterizations of Pt-S-CNT, Pt-N-CNT and Pt-C-CNT, respectively. The edges of multiple rolled graphene layers are clearly resolved, and isolated Pt single atoms are identified as the bright dots due to their higher Z contrast compared to the neighboring C/N/S sites. This atomically dispersed feature has been confirmed by Fourier-transformed extended X-ray absorption fine structure spectra (FT-EXAFS) at the Pt $L_3$-edge (Fig. 1d). Pt foil exhibits a typical first shell Pt–Pt pair at ~ 2.55 Å, while Pt–O interaction in $PtO_2$ locates at 1.93 Å, neither of them is observed on the as-prepared catalysts. The major R-space feature is noted as 1.90 Å for Pt-S-CNT, 2.36 Å for Pt-C-CNT and 1.60 Å for Pt-N-CNT, probably arisen from the characteristic bonding of Pt-S, Pt-N and Pt-C, respectively[33,36] (EXAFS fitting results show in Supplementary Fig. 3).

The composition of Pt-X-CNT and relevant Pt valence state are further probed by ex situ X-ray photoelectron spectroscopy (XPS). XPS survey spectra in Supplementary Fig. 4 depict the componential information, for which a similar Pt content of ~ 0.03 at% (ca. 0.7 wt%) is observed on all three samples. For Pt-N-CNT, core-level deconvoluted XPS spectrum on N 1 s region is plotted in Supplementary Fig. 5, with five different coordination structures pinpointed as pyridine-N (398.5 eV, 23.55%), pyrrole-N (400.4 eV, 33.81%), graphitic-N (401.6 eV, 16.40%), oxidized-N (403.1 eV, 10.63%) and N-Pt bonding (399.2 eV, 15.62%)[37,44]. For Pt-S-CNT, the S 2p spectrum shows two major peaks located at 164.2 eV and 165.9 eV (Supplementary Fig. 6), attributable to C-S-C (90.37%) and –$SO_x$ (9.63%) coordination, respectively[45]. Figure 1e plots the core-level XPS spectra for Pt 4 f region, in which the Pt $4f_{7/2}$ binding energy increases in the order of Pt-C-CNT (71.97 eV) < Pt-N-CNT (72.35 eV) < Pt-S-CNT (72.52 eV). Notably, all these values are higher than that of metallic Pt (71.2 eV)[31], indicating a partial charge transfer from Pt central atom to neighboring metalloid dopant and in good agreement with previous SACs report[43,46]. This characteristic of partial

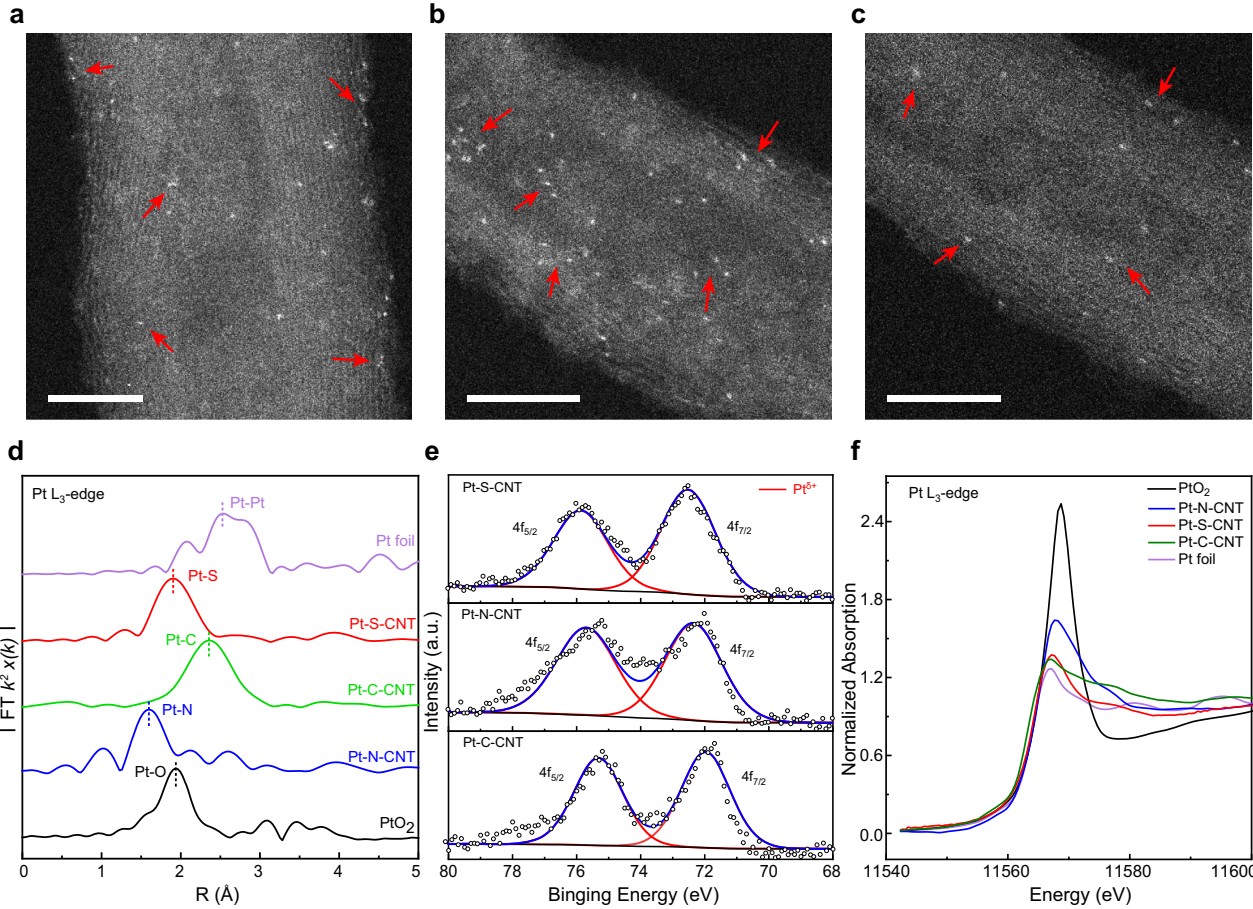

**Fig. 1 Characterizations of Pt-X-CNT catalysts.** HAADF-STEM images of **a** Pt-S-CNT, **b** Pt-N-CNT, and **c** Pt-C-CNT, the bright dots (as marked by red arrows) represent typical Pt single atoms. Scale bars, 5 nm. **d** $k^2$-weighed Pt $L_3$-edge FT-EXAFS spectra for as-prepared Pt-X-CNT in the $R$ space, **e** core-level XPS spectra for Pt 4 f region, and **f** corresponding XANES spectra at Pt $L_3$-edge in comparison to Pt foil and $PtO_2$ references.

depleted free electrons from Pt valence band has been further reinforced by the X-ray absorption near-edge structure (XANES) spectra shown in Fig. 1f, in which the white line peaks for Pt SACs locate in between Pt foil and $PtO_2$.

The ORR performance was evaluated 0.1 M $HClO_4$ electrolyte on Pt-X-CNT catalysts cast rotating ring disk electrode (RRDE). Prior to RRDE measurements, the collection coefficient of Pt ring electrode was pre-calibrated as 37.09% through the redox reaction of $[Fe(CN)_6]^{4-}/[Fe(CN)_6]^{3-}$ (Supplementary Fig. 7)[47]. Figure 2a shows the polarization curve and corresponding $H_2O_2$ partial current recorded on Pt-X-CNT in comparison to bare N-CNT, relevant $H_2O_2$ selectivity and electron transfer number ($n$) as a function of applied potential are plotted in Fig. 2b. Other linear sweep voltammograms of S-CNT and defective CNT substrates are depicted in Supplementary Fig. 8. At the absence of Pt, N-CNT demonstrates the earliest onset potential of 0.42 V (defined as the potential delivering 0.1 mA cm$^{-2}$ $H_2O_2$ partial current density) and maintains 47.9–59.1% $H_2O_2$ selectivity throughout the investigated potential window, which is comparable to literature reports[33,37,38,48,49]. Defective CNT prepared from hydrothermal treatment[38] shows an even higher $H_2O_2$ selectivity up to ∼ 75%, which could be attributable to carbon defect sites[50] and/or oxygen functional groups modification[51].

Besides, taken Pt-S-CNT, S-CNT and reported O-CNT[51] for example, the recorded current densities within kinetics-controlled regime as well as the overall $H_2O_2$ selectivity are significantly improved as compared to Pt-free counterparts, indicating that Pt central atoms rather than the metalloid doped C sites or the

O-dopants serve as the main active center for peroxide generation (Supplementary Fig. 9). During the negative-going potential sweep, Pt-S-CNT tends to catalyze the $O_2$-to-$H_2O_2$ conversion, delivering a maximum $H_2O_2$ selectivity above 88% and an early onset potential of 0.51 V with optimized preparation conditions (Supplementary Fig. 10). This $2e^-$ ORR performance is among the first echelon of reported values in acidic media, as summarized in Fig. 2c and Supplementary Table 1. An even higher kinetic current density may be expected from a higher concentration of isolated Pt-$S_x$ moieties like Pt/HSC and Pt$_1$-$CuS_x$[31,33]. For Pt-N-CNT, the electrocatalytic ORR pathway is found to be affected by the annealing temperature. Supplementary Fig. 11 shows the $H_2O_2$ generation profile as a function of pyrolysis temperature, in which the 800 °C annealed Pt-N-CNT delivers a highest $H_2O_2$ selectivity up to 72.5% and an early onset potential of 0.62 V as plotted in Fig. 2a and b. As to Pt-C-CNT, the $4e^-$ ORR pathway dominates at small overpotential region, while $H_2O_2$ selectivity increases with increasing overpotential. By increasing Pt loading from 0.7 to 4.3 µg cm$^{-2}$, an even higher diffusion limited current density of 5.7 mA cm$^{-2}$ can be achieved (Supplementary Fig. 12), in good agreement with earlier report[38].

To further evaluate the intrinsic ORR activity and selectivity, electrochemical active surface area and relevant Pt site density over Pt-X-CNT are quantified by CO stripping measurements[52,53], assuming a 1:1 molar ratio for $CO_{ad}$ to the adsorbed Pt sites given its predominant linear adsorption configuration on Pt surface[54] (Supplementary Fig. 13 and Supplementary Table 3). As shown in Supplementary Fig. 14, the determined $TOFs_{H_2O_2}$ on Pt-S-CNT and

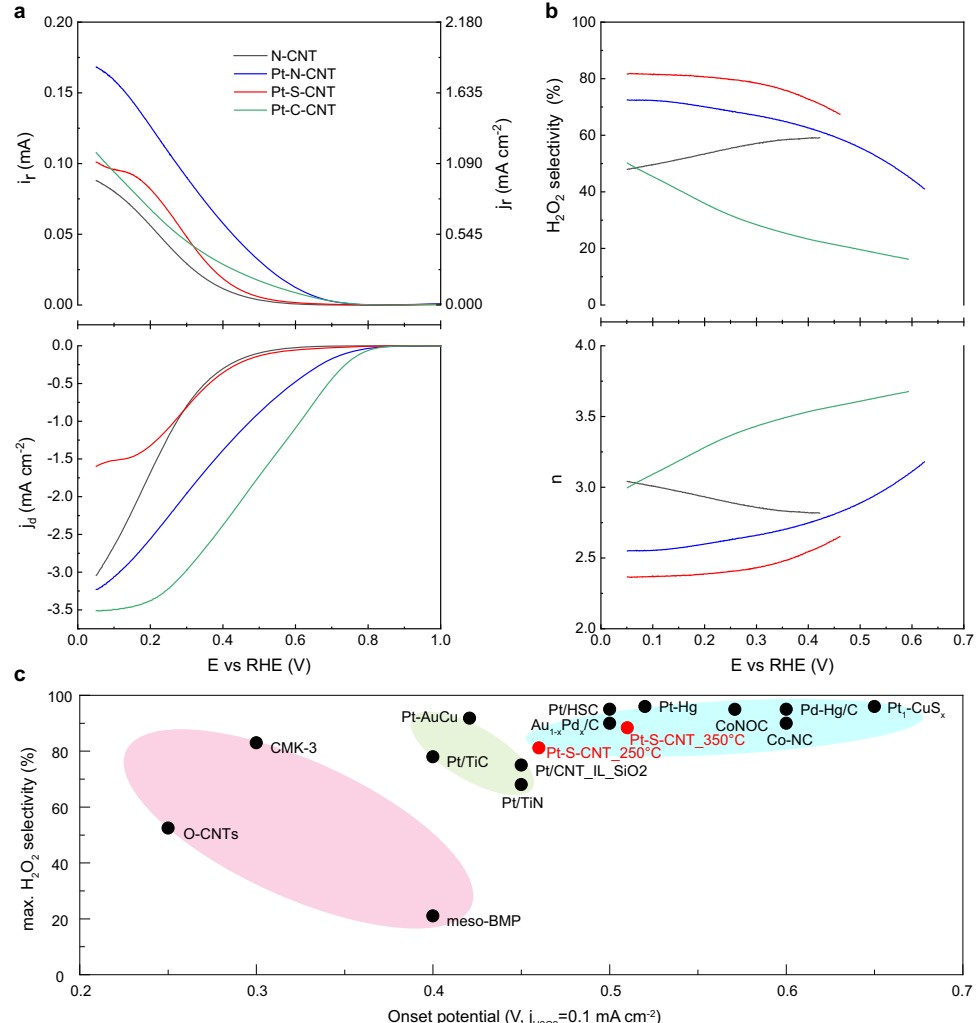

**Fig. 2 ORR performance of Pt-X-CNT catalysts cast RRDE in 0.1 M HClO₄. a** Linear sweep voltammetry (LSV) of Pt-N-CNT (blue), Pt-S-CNT (red), Pt-C-CNT (green) and N-CNT reference recorded at 1600 rpm and a scan rate of 5 mV s⁻¹, together with the detected H₂O₂ currents on the ring electrode (upper panel) at a fixed potential of 1.2 V vs. RHE. The catalyst loading was fixed at 0.1 mg cm⁻². **b** Calculated H₂O₂ selectivity and electron transfer number (*n*) during LSV scan. H₂O₂ selectivity and *n* were plotted from the onset potential that reached 0.1 mA cm⁻² H₂O₂ partial current density. **c** Performance map for the electrocatalytic O₂-to-H₂O₂ conversion in acidic electrolyte, the detailed sample information could be found in Supplementary Table 1.

Pt-N-CNT are significantly higher than TOFs$_{H2O}$, while a reverse trend is observed on Pt-C-CNT. Based on the ratio of TOF$_{H2O2}$/TOF$_{H2O}$, an increasing 2e⁻/4e⁻ ORR pathway selectivity is noted as the order of Pt-C-CNT < Pt-N-CNT < Pt-S-CNT, which is in good agreement with RRDE results. Noteworthy, by switching the coordination motif from Pt-C to Pt-N and Pt-S, the H₂O₂ selectivity at a given potential of 0.4 V increases from 23.3% to 62.5% and 81.4% and a TOF$_{H2O2}$/TOF$_{H2O}$ increases from 0.30 to 1.67 and 2.67, respectively, highlighting the role of local coordination environment in determining the oxygen reduction pathway.

**Theoretical calculations**. The above electrochemical measurements suggest ORR product selectivity largely depends on the Pt coordination environment. Herein, we performed density functional theory (DFT) calculations on Pt-X-C moieties to shed light on the origin of observed pathway tuning. In line with earlier report, a two-dimensional graphene structure is deployed to model the CNT support[43], and Fig. 3a shows the simplified coordination of Pt-S₄, Pt-N₄ and Pt-C₄ motifs in comparison to Pt(111) surface upon the adsorption of *OOH. The formation

energy (ΔE$_f$) is ca. −0.16, −2.09 and −1.61 eV for each motif, suggestive a stable coordination configuration.

Prior to ORR simulations, a simple electrochemical hydrogen evolution (HER) measurement coupled with thermodynamic analysis have been carried out as a probe reaction to examine the proposed models. Supplementary Fig. 15 shows the voltammograms of Pt-X-CNT as recorded in 0.1 M Ar-saturated HClO₄ electrolyte from −0.05 to 1.05 V vs. RHE. Only hydrogen evolution and oxidation peaks are observed during the cyclic voltammetry, the absence of Pt redox peaks at high potential region confirms the highly dispersed feature of prepared Pt-X-CNT catalysts. The Gibbs free energy of adsorbed *H (ΔG$_H$, Supplementary Fig. 16) is used as the descriptor for HER activity, for which the absolute value decreases in the order of Pt-S₄ > Pt-N₄ > Pt-C₄, in good harmony with our experimental observation thus confirming the validity of proposed configurations.

Fig. 3b schematically compares associative and dissociative O₂ reduction pathway[55,56], and the free energy diagram for each reaction step on different moieties, including Pt(111) model surface, is calculated at 0.4 V and plotted in Fig. 3c–e. The hydrogeneration of *OOH intermediate is found to be the critical

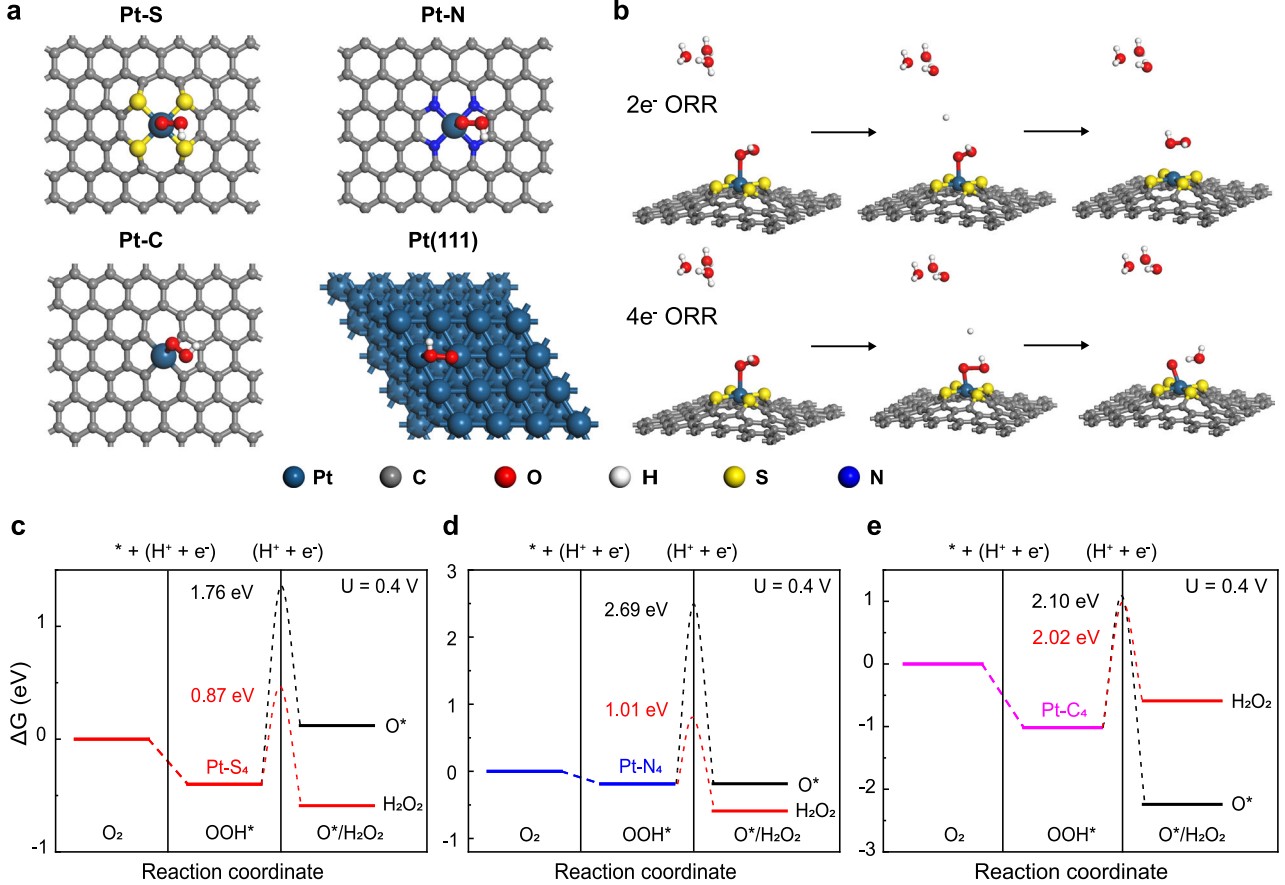

**Fig. 3 DFT calculations of the ORR selectivity on different Pt-X-C moieties. a** Examined configurations for Pt single atom coordinated in two-dimensional carbon matrix with different metalloid doping, *OOH adsorption is preferable on central Pt sites over neighboring S/N/C atoms. **b** Illustration of oxygenated intermediates adsorption taking Pt-S$_4$ as an example, and relevant free energy diagrams for 2e$^-$ (red) or 4e$^-$ (black) ORR pathway on **c** Pt-S$_4$, **d** Pt-N$_4$ and **e** Pt-C$_4$ motif. Insert numbers represent the kinetic barriers for *OOH to H$_2$O$_2$ or *O, as computed by the Climbing Image Nudged Elastic Band (CI-NEB) approach.

knob branching ORR pathway[4]. Similar to that on Pt(111), a dissociate reduction of *OOH to *O is found to be energetically favored on Pt-C$_4$, while the associate reduction of *OOH to H$_2$O$_2$ is favored on Pt-S$_4$ and Pt-N$_4$ moieties. The energetic differences between 2e$^-$ and 4e$^-$ ORR pathways, i.e., $\Delta G_{assoc}$–$\Delta G_{dissoc}$, are comparatively depicted in Supplementary Fig. 17, in which a most energetic favorable O$_2$-to-H$_2$O$_2$ conversion of ca. −0.71 eV is noted on Pt-S$_4$, followed by −0.41 eV on Pt-N$_4$ and unfavorable conversion of +1.65 eV on Pt-C$_4$ and +2.67 eV on Pt(111). We also compare this energetic difference on Pt single atom site *versus* that on Pt cluster using Pt$_1$-S$_4$-C and Pt$_6$-S$_4$-C as a representative[57], i.e., −0.71 eV for the former and +2.83 eV for the latter (Supplementary Fig. 18), which in turn verifies the isolated Pt sites feature from theoretical side. Besides, we consider both the Pt-C$_4$ (Fig. 3a) and the Pt-C$_3$ coordination (Supplementary Fig. 19) for Pt-C-CNT, as derived from EXAFS fitting results, the 4e$^-$ pathway is found to be more energetic favorable on both Pt-C motifs. Despite these thermodynamic simulations, we further compare the kinetic barriers for both 4e$^-$ and 2e$^-$ ORR pathways over different Pt-coordinated moieties. The results show that Pt-S$_4$ exhibits the lowest kinetic energy barrier for 2e$^-$ pathway (0.87 eV), followed by Pt-N$_4$ of 1.01 eV and Pt-C$_4$ of 2.02 eV. The simplified analysis on thermodynamics and reaction kinetics actually matches well with the trend of experimentally determined H$_2$O$_2$ selectivity, reinforcing the Pt local coordination environment effect on fine tuning the binding strength of *OOH intermediate and thus altering the oxygen reduction pathway.

**H$_2$O$_2$ production from bulk electrolysis.** Decentralized H$_2$O$_2$ production with tunable aqueous concentration is ideal for onsite applications[58,59]. Herein, we cast the above pinpointed Pt-S-CNT catalyst onto a $1 \times 1$ cm$^2$ carbon fiber paper electrode and further investigate its potential toward H$_2$O$_2$ generation from bulk electrolysis. The H-type reactor is schematically shown in Fig. 4a, where the cathodic catalyst loading of Pt-S-CNT is fixed at 0.5 mg cm$^{-2}$ (ca. 3.5 µg$_{Pt}$ cm$^{-2}$). A homogeneous dispersion of Pt-S-CNT over carbon fibers substrate can be seen from the scanning electron microscopic image (SEM, Fig. 4b) and the MicroCT 3D tomography (Supplementary Fig. 20). The ORR activity for as-prepared catalyst and the one post long-term electrolysis are evaluated in 25 mL of 0.1 M O$_2$-saturated HClO$_4$, and corresponding LSV curves are plotted in Fig. 4c. For as-prepared catalyst, the ORR onset potential locates at ~ 0.6 V, and the current density reaches ~ 10 mA cm$^{-2}$ at 0.05 V vs RHE. Long-term electrolysis is then carried out at a constant current density of 10 mA cm$^{-2}$. 1 mL of electrolyte solution was extracted from cathodic chamber every 30 min during the electrolysis, and the content of H$_2$O$_2$ was determined by potassium permanganate titration[60]. Figure 4d shows a negligible potential decay to deliver the constant current of 10 mA, and Supplementary Fig. 21 depicts the inductive coupled plasma - mass spectroscopic results of ~ 0.29 at.% Pt leaching from Pt-S-CNT into bulk electrolyte (ICP-MS, ca. 0.4 ppb Pt leaching in total). Moreover, ~ 70% selectivity of O$_2$-to-H$_2$O$_2$ conversion is maintained during the 500-min electrolysis, demonstrating a stable peroxide generation

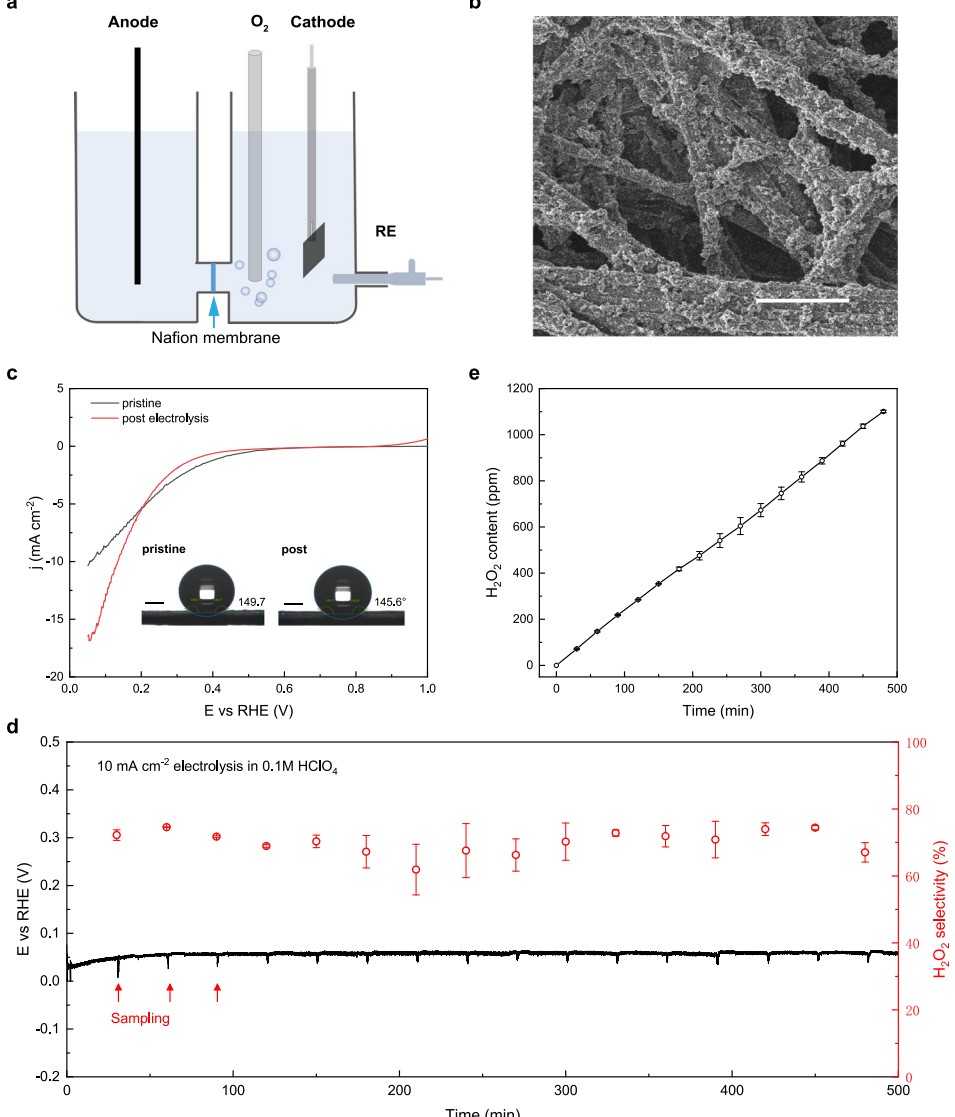

**Fig. 4 Application of Pt-S-CNT catalyst in a H-cell electrolysis. a** Schematic of a home-made H-type electrolyzer. **b** SEM image of Pt-S-CNT cast carbon fiber paper as working electrode, scale bar: 500 nm. **c** LSV curves recorded on either pristine or post-electrolysis working electrode within 0.1 M $O_2$-saturated $HClO_4$ at a scan rate of 5 mV s$^{-1}$, inset shows the contact angles of a same working electrode prior to and post 500-min electrolysis at the scale bar of 1 mm. **d** The ORR stability test of working electrode potential and $H_2O_2$ selectivity under a fixed current density of 10 mA cm$^{-2}$ for 500 min continuous electrolysis. **e** The accumulated $H_2O_2$ concentration during the stability measurement. The error bars represent two independent samples.

performance over Pt-S-CNT. After this long-term operation, the ORR current density is noted to exceed 15 mA cm$^{-2}$ (Fig. 4c) at 0.05 V but at the sacrifice of onset potential, which may arise from the gradually increased hydrophilicity of the electrode surface as can be seen from the decreased contact angle. More than 1100 ppm $H_2O_2$ was accumulated during this electrolysis (Fig. 4e), which can be further scaled up by integrating with gas diffusion electrodes into solid-state electrolyzers and flow cell reactors[61,62].

**Effect of Pt sites density on $H_2O_2$ generation**. In addition to the above coordination environment tuning, increasing metal loading[63,64] and/or peroxide diffusion paths[65] have also been demonstrated to improve the selectivity of $O_2$-to-$H_2O$ conversion. Along this line, we adapted a series of Pt-N-CNT catalysts with different Pt loadings as a stereotype to investigate the isolated Pt site density effect on ORR pathway tuning, since M-N-C

moiety[66] is generally considered as the most promising alternative to Pt in 4e$^{-}$ ORR.

The amount of Pt precursor solution was adjusted from 200 to 2000 μL, namely 200-, 400-, 800- and 2000-Pt-N-CNT, corresponding to the Pt loading of 0.7 wt.%, 1.4 wt.%, 2.8 wt.% and 7.0 wt.%, respectively. An identical CNT structure is found on all the four catalysts as shown in Fig. 5a of bright field TEM characterization. No Pt nanoparticles were observed at relatively low Pt loading of 200-, 400- and 800-Pt-N-CNTs, indicating a highly dispersed feature of Pt sites in these samples as due to the strong Pt-N interaction[35,37]. For 2000-Pt-N-CNT, Pt aggregates at a diameter of ~10 nm were found as black spots on CNT substrate. XPS survey spectra over these Pt-N-CNTs are plotted in Supplementary Fig. 22, in which only C, O, N, Pt elements were identified. Besides, the content of surface Pt species increases with increasing Pt precursor amount from 200 to 800 μL but decreases for 2000-Pt-N-CNT probably arisen from the lower percentage of (sub)surface atoms in nanoparticles as compared to

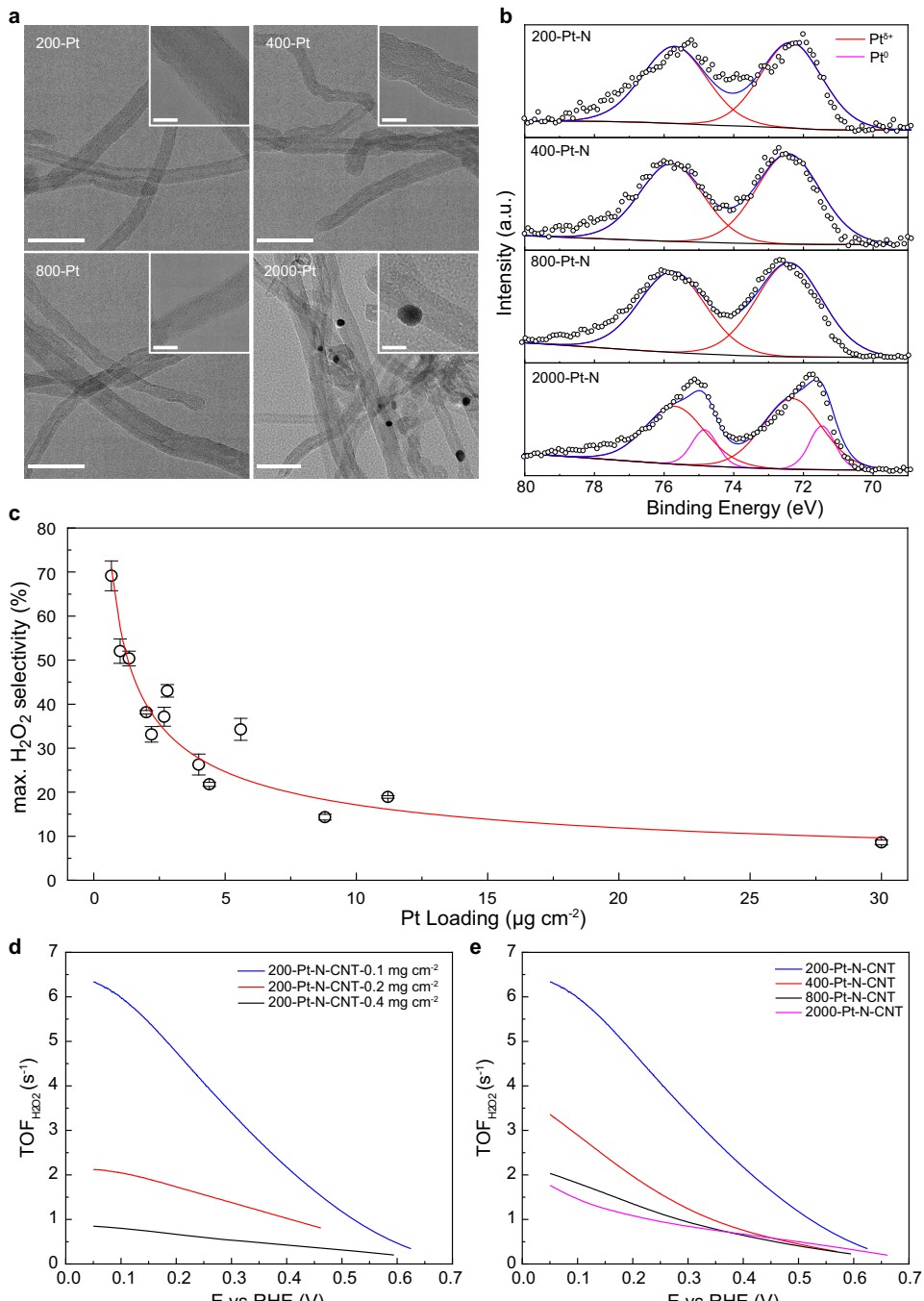

**Fig. 5 Pt loading effect on H₂O₂ selectivity. a** TEM images of 200-, 400-, 800- and 2000-Pt-N-CNT. Scale bars, 50 nm. Inserts are the zoom-in view of certain carbon nanotubes at a fixed scale bar of 10 nm. **b** Core-level XPS spectra for Pt 4 f region. **c** The maximum H₂O₂ selectivity as a function of Pt loading. The rightmost data point was measured on commercial TKK 30 wt.% Pt/C catalyst and plotted with actual Pt weight loading, while all other data on Pt-N-CNT were plotted using determined Pt content from XPS measurements. **d** Calculated TOFs of 200-Pt-N-CNT with different catalyst loading, and **e** calculated TOFs of *n*-Pt-N-CNT with different Pt site density but a fixed catalyst loading of 0.1 mg cm⁻².

single atom catalyst. The core level XPS on Pt 4 f region are plotted in Fig. 5b, a sole characteristic of Pt$^{\delta+}$ peak located at 72.35 eV is noted on 200-, 400- and 800-Pt-N-CNTs while both Pt$^0$ (71.46 eV) and Pt$^{\delta+}$ peaks are resolved on 2000-Pt-N-CNT, in good agreement with above microscopic characterization.

To probe the Pt sites density effect on H₂O₂ selectivity, 2 variables are considered in our RRDE measurements, i.e., one with fixed Pt-N-CNT catalyst loading of 0.1 mg cm⁻² but different Pt content from 0.7 wt.% to 7.0 wt.% and the other one with gradually increased catalyst loading but a fixed Pt

content. As shown in Supplementary Fig. 23, the maximum H₂O₂ selectivity for all *n*-Pt-N-CNTs significantly decreases with increasing catalyst loading from 0.1 to 0.4 mg cm⁻². Increasing catalyst loading could lead to a thicker catalyst layer and thus a longer diffusion path for H₂O₂ product. During its diffusion process, a further reduction of H₂O₂ into H₂O is expected which reduces the final peroxide selectivity[64,67]. Moreover, we normalize the surface Pt sites density from XPS survey results and plot the Pt loading dependence of maximum H₂O₂ selectivity in Fig. 5c. By increasing the Pt active sites density, the 2e⁻ ORR pathway

could be successfully shifted toward the 4e⁻ pathway, with the $H_2O_2$ selectivity decreasing from 70 to 20% and the electron transfer number increasing from 2.55 to 3.62. Figure 5d plots the overall catalyst loading effect on $TOF_{H_2O_2}$ over 200-Pt-N-CNT, i.e., the thinner diffusion layer thickness, the higher $H_2O_2$ generation rate per Pt site. Similarly, at a fixed catalyst loading of 0.1 mg cm⁻², $TOF_{H_2O_2}$ decreases with increasing electrochemical active Pt site density from 0.29 wt.% to 0.51 wt.% as derived from CO stripping measurements (Fig. 5e). Taken together the Pt loading dependence of apparent $H_2O_2$ selectivity and the intrinsic TOF quantification, it can be rationalized that the densely distributed Pt sites are beneficial for the total reduction of $O_2$ into $H_2O$ rather than $H_2O_2$ generation.

## Discussion

Through a combined experimental and theoretical approach, we demonstrate that manipulating the local coordination environment of Pt single atoms could effectively regulate the ORR pathway. Pt-S-C moiety is inclined to the 2e⁻ pathway of $O_2$-to-$H_2O_2$ conversion while Pt-C moiety is mostly favorable to the totally reduction of $O_2$ into $H_2O$. In addition, the peroxide selectivity is further correlated with the Pt active sites density, with a tunable electron transfer number from 2.55 to 3.62 successfully demonstrated on Pt-N-C prototype. These understandings will aid to the development of more efficient and selective Pt single atom catalysts for both 4e⁻ ORR in fuel cell applications and 2e⁻ ORR in green peroxide synthesis. Given the wide application scenarios of Pt and the abundant coordination motif structures, the present approach sheds light on the design of more delicate (electro-)catalyst, which is of great importance to both fundamental surface catalysis studies and practical energy conversion process.

## Methods

**Material synthesis and characterization**. The Pt-X-CNT catalyst (X = N, S, C) was prepared by the impregnation-reduction method. In a typical Pt-N-CNT synthesis, $K_2PtCl_4$ (AR, Sinopharm) precursor was firstly dissolved in Millipore water (18.2 MΩ·cm) to prepare a 3 mg mL⁻¹ stock solution. A carbon suspension was then prepared by mixing 40 mg of multi-walled carbon nanotubes (Carbon Nanotubes Plus GCM389, used as received) with 25 mL of Millipore water, and tip sonicated for 30 min. Thereafter, 200 μL of Pt stock solution was added dropwise to the CNT suspension under vigorous stirring for a 0.7 wt.% Pt target loading and got quickly frozen in liquid nitrogen. The amount of Pt stock solution was adjusted accordingly to regulate the different Pt loadings. The prepared $K_2PtCl_4$/CNT powder was grinded with urea (AR, Sinopharm) at a mass ratio of 1:10, then heated up to 800 °C within 60 min in a tube furnace under an Ar (99.999%, Shanghai Coogee) flow of 100 sccm, and kept at the same temperature for another 60 min before cooling down to room temperature[37]. The preparation method of Pt-S-CNT and Pt-C-CNT is similar to that of Pt-N-CNT, except the pretreatment of CNT. To prepare S-doped CNT in Pt-S-CNT, CNT and diphenyl disulfide (PDS, 99%, Macklin) were mixed at a mass ratio of 1:4, heated up to 900 °C within 30 min under 100 sccm Ar flow in a tube furnace, and kept at the same temperature for another 30 min before cooling down[45]. The $K_2PtCl_4$/S-CNT powder was heated to 250 °C with 100 sccm Ar within 30 min and kept for another 120 min before cooling down to obtain the Pt-S-CNT. As to Pt-C-CNT, 50 mg of CNT were uniformly dispersed in 12.5 mL of absolute ethanol (AR, Sinopharm) + 12.5 mL of Millipore water + 250 μL of $H_2O_2$ (GR, Sinopharm), then got transferred to a 100-mL autoclave for reaction at 200 °C for 6 h. The obtained defective CNT[38] was centrifuged and washed intensively with Millipore water for 2 times. The $K_2PtCl_4$/defective-CNT powder was heated to 900 °C within 60 min under 100 sccm Ar flow, kept at 900 °C for another 60 min and then cooled to room temperature to obtain the Pt-C-CNT.

The SEM images were taken on a FEI Sirion 200 Field-Emission scanning electron microscopy, using an electron beam energy of 5 kV and a spot size of 3.0 nm with magnification ranging from 5 to 80 k. A bright field transmission electron microscope, Talos F200X G2, was used to characterize the morphology of Pt-X-CNT. Dark field STEM characterizations were carried out using a JEM ARM200F aberration-corrected transmission electron microscope under 200 kV. Drift correction was applied during acquisition. X-ray photoelectron spectroscopy was obtained with a Kratos AXIS Ultra DLD spectrometer, using a monochromatic Al Kα radiation (1486.6 eV) and a low energy flood gun as neutralizer. The binding energies were calibrated by referencing to C 1s peak at 284.8 eV. Casa XPS program was employed for surface component analysis, with detailed fitting

parameters tabulated in Supplementary Table 2. X-ray absorption spectroscopy (XAS) at Pt $L_3$-edge of Pt-S-CNT and Pt-N-CNT was measured at beamline 7-BM (QAS) of National Synchrotron Light Source II (NSLSI-II) in Brookhaven National Laboratory (BNL), and Pt-N-CNT was measured at the beamline BL14W1 of Shanghai Synchrotron Radiation Facility (XAFS station, SSRF). The data were collected in the fluorescence mode and the reference spectrum of Pt metal foil was also collected during the measurement for energy calibration. XANES and EXAFS data were processed and analyzed using the Athena and Artemis software package[68]. MicroCT was performed on a Zeiss Xradia 520 Versa X-ray microscopy. The distances of the sample to the X-ray source (12.53 mm) or to the X-ray detector (72.09 mm) result in a voxel (volume pixel) size of 1 μm. The field of view was approximately 1012.3 μm × 1012.3 μm, and the reconstruction of MicroCT data was conducted using the TXM Reconstructor software (Xradia). XRD spectra were recorded on a Rigaku Mini Flex 600 spectrometer using a Cu Kα radiation (40 kV, 15 mA) at a scan rate of 0.02° per step and a holding time of 4 s per step. ICP-MS was measured on a Thermo Scientific iCAP-Q spectrometer to quantify the potential Pt leaching ratio during electrolysis with a detection limit of 0.001 $ppb_{Pt}$.

**Electrochemical measurements**. 0.1 M $HClO_4$ was used as the electrolyte, which was obtained by dissolving 14.315 g of concentrated $HClO_4$ (70%, Sigma) in 1 L Millipore water. The electrochemical measurements were run in a standard three-pathway system, with signals recorded by a CHI 760e working station. An RRDE component (AFE7R9, Pine Instruments) consisting of a glassy carbon rotating disk electrode (Φ = 5.61 mm) and a Pt ring (Φ = 6.25 mm) was used, with a theoretical collection efficiency of 37%. Experimentally, the apparent collection efficiency (N) was determined to be 37.1% by using the ferrocyanide/ferrocyanide half-reaction system at rotation rate between 400 and 2025 rpm. A high-purity graphite rod (99.995%, Sigma-Aldrich) and a fresh prepared reversible hydrogen electrode (RHE) was used as the counter and the reference electrode, respectively. To prepare RHE, a capillary tube sealed with platinum wire was first filled with 0.1 M $HClO_4$, and then a part of hydrogen was generated through electrolysis, forming a saturated Pt/$H_2$ interface. To prepare the catalyst cast RRDE, typically 3.3 mg of Pt-X-CNT was firstly mixed with 0.5 mL of Millipore water, 0.5 mL of isopropanol (AR, Sinopharm) and 15 μL of Nafion 117 solution (5 wt.%, Sigma-Aldrich), ultra-sonicated for 30 min till a homogeneous catalyst ink. Then 7.5 μL of the catalyst ink was pipetted onto mirror-polished carbon disk (0.247 cm², ca. 0.1 mg cm⁻² mass loading). The hydrogen peroxide selectivity was calculated using the following equation: $H_2O_2(\%) = 200 \times \frac{I_{Ring}/N}{|I_{Disk}| + I_{Ring}/N}$, the number of electrons transferred on the disk electrode in the ORR process n through the equation: $n = 4 \times \frac{|I_{Disk}|}{|I_{Disk}| + I_{Ring}/N}$, where $I_{Ring}$ is the ring current, $I_{Disk}$ is the disk current, and N is the determined collection efficiency.

The bulk electrolysis of ORR was performed in a customized H-type cell, Pt-S-CNT was sprayed onto a 1 × 1 cm² carbon fiber paper (HCP010N, Hesen) as the cathode (catalyst loading of 0.5 mg cm⁻²), and the high-purity graphite rod was used as the anode, with the two chambers separated by Nafion 212 cation exchange membrane. 1 mL of the cathodic electrolyte was extracted every 30 min during the electrolysis, and the generated $H_2O_2$ concentration was quantified by potassium permanganate (0.02 M $KMnO_4$ solution, AR, Sinopharm) titration method[60] with a detection limit of ca. 8.5 ppm (Supplementary Fig. 24):

$$2MnO_4^- + 5H_2O_2 + 6H^+ \rightarrow 6Mn^{2+} + 5O_2 + 8H_2O \quad (3)$$

Electrochemical CO stripping measurements were carried out on Pt-X-CNT cast RDE working electrode, starting with bubbling CO (99.9%, Air Liquide) in 0.1 M $HClO_4$ for 20 min at 0.25 V vs. RHE. Then, the dissolved CO was removed from the electrolyte by Ar purging for 60 min while maintaining the electrode potential at 0.25 V. Finally, we recorded the CO stripping voltammograms by scanning the potential from 0.25 to 1.30 V at 5 mV s⁻¹.

**Computational details**. The density functional theory (DFT) calculation is performed by Vienna ab initio Simulation Program (VASP)[69–72]. The PBE functional with projector augmented wave pseudo-potential is used to calculate all models[73,74]. A Gaussian smearing technique is used with a smearing parameter of $k_BT = 0.1$ eV for the fractional occupation of the one-electron energy levels to accelerate SCF convergence and calculated energies are extrapolated to $k_BT = 0$ eV.

The surface of CNT is described by the monolayer graphite model, of which lattice parameters are α = 14.760 Å, β = 17.0434 Å, c = 20.0 Å. Carbon-based slab calculations are sampled by a Monkhorst-Pack k-point net of 5 × 4 × 1 with cutoff energy of 500 eV. (111) facet of Pt consists of 5 slabs and sampled by a Monkhorst-Pack k-point net of 5 × 5 × 1 with cutoff energy of 500 eV. Molecule calculations are sampled by Gamma point with cutoff energy of 400 eV.

The energy of (H⁺ + e⁻) is defined as two times of that of $H_2$ on the standard condition, i.e., U = 0 and pH=0[75,76]. The free energy was calculated as:

$$G = E + ZPE - TS - neU \quad (4)$$

where E is the DFT energy of the model, ZPE is the zero-point energy which is calculated by $\Sigma(h\nu i/2)$ (h is the Planck constant and $\nu i$ is the vibrational frequency), T is the temperature (298.15 K), S is the entropy of the structure which is given by

vibrational frequency, $n$ is the number of electrons transferred in elementary reaction, $e$ is the charge constant and $U$ is the potential[75,76]. The energy of $H_2O$ is calculated by $G_{H2O(l)}=G_{H2O(g)} + RT \times In(P/P0)$, where $G_{H2O(g)}$ is given by free energy as above, $R$ is gas constant with $P_0 = 1$ bar and $P = 0.035$ bar[75,76]. The free energy of $O_2$ is calculated according to the thermodynamic energy (4.92 eV) released by the reaction of $2H_2 + O_2 \rightarrow 2H_2O$. The thermodynamic energy released by the reaction of $H_2 + O_2 \rightarrow H_2O_2$ is $-1.39$ eV. To testify if any curvature effect on the calculated ORR free energy diagrams, a representative 2D Pt-$S_4$-C motif is plotted against 3D Pt-$S_4$-CNT model as shown in Supplementary Fig. 25. Nearly identical *OOH binding strength is observed on the two cases, suggestive a minor curvature effect on $H_2O_2$ generation.

## Data availability

The data generated in this study are provided in the Supplementary Information. Source data are provided with this paper.

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

## Acknowledgements
This work was supported by the National Natural Science Foundation of China (NSFC, No. 22002088), the Shanghai Sailing Program (No. 20YF1420500) and the Oceanic Interdisciplinary Program of Shanghai Jiao Tong University (No. SL2020MS007). F.J. acknowledges the support from NSFC under the Grant No. 21802095. This research used resources at beamline 7-BM (QAS) of the National Synchrotron Light Source II, a U.S. Department of Energy (DOE) Office of Science User Facility operated for the DOE Office of Science by Brookhaven National Laboratory under Contract No. DE-SC0012704. The authors would like to thank BL14W1 in Shanghai Synchrotron Radiation Facility (SSRF) for providing the beam time to perform part of XAFS measurements.

## Author contributions
K.J. conceptualized the project. Ji.Z. and K.J. developed and performed catalyst synthesis, conducted the electrochemical tests and the related data processing. Ji.Z. performed materials characterization with the help of K.Y., Z.L. and X.Z. XAFS experiments and relevant analysis were carried out by F.J., L.M. and Z.S. and X.W. C.F., S.S. and Ju.Z. performed the DFT simulations. Ji.Z., C.F. and K.J. wrote the paper. K.J. and Ju.Z. supervised the project. All authors discussed the results and commented on the paper.

## Competing interests
The authors declare no competing interests.
