## [Peer Review File · Nature Communications]

Title: Manipulating the oxygen reduction reaction pathway on Pt-coordinated motifsREVIEWER COMMENTS

Reviewer #1 (Remarks to the Author):

In this work, the authors prepared a series of Pt-X-C single atom catalysts and investigated the influence of local coordination environment on acidic oxygen reduction reaction pathway tuning. Isolated Pt-S motif is identified as the most selective one toward 2e pathway while 4e pathway is predominant on Pt-C motif. The experimental RRDE and bulk electrolysis researches combined with energetic analysis provide precise answer as to how the neighboring dopants influencing the reaction pathway, which is largely controversial in recent literatures. This present methodology and electrocatalytic results could make a great significance to the active motifs engineering in single atom catalysts as well as to shed light on the structure-activity relationship. I would suggest its publication at Nat. Commun. after addressing the comments below:

1. For XPS spectra fitting, detailed parameters like the peak position, FWHM should be tabulated.
2. In Supplementary Figure 12, the authors considered the kinetic barrier of Pt-S-C motif for either 4e or 2e pathway, for which the latter is significantly lower than the former, suggestive both thermodynamically and kinetically favorable O₂-to-H₂O₂ conversion. The authors are encouraged to supplement a similar kinetic analysis on Pt-C and Pt-N-C coordination structure.
3. Regarding the H₂O₂ quantification from bulk electrolysis, what's the detection limit for this KMnO₄ titration method, i.e., 1 ppm, 10 ppm?
4. Open question, since O-doped carbon is widely reported to effectively convert O₂ into H₂O₂ in electrochemistry, how to differentiate the contribution of Pt active motif from O-doped CNT substrate?

Reviewer #2 (Remarks to the Author):

This manuscript describes the synthesis of Pt-(C, S, N)-CNT catalysts and their catalytic performance for 4e- pathway or 2e- pathway ORR. In terms of catalytic performance for ORR, the reported results are inferior to many recently published studies. The highest selectivity towards H₂O₂ is below 80%, much lower than many recent studies on Pt catalysts, oxidized carbon catalysts, and transition metal single-atom catalysts. Thus, the potential novelty of the current work is on the atomic structural tuning of single-atom sites and the understanding of their catalytic behaviors. Although different structures of M-N-C sites have been studied in several previous studies, the specific Pt-(C, S, N)-CNT comparison has not been reported before.

The authors used high-temperature thermal treatment methods to prepare Pt-(C, S, N)-CNT catalysts. They also provided XAS, XPS, and TEM images to prove that only single atom sites were obtained. XAS provides the average coordination environment of bulk samples. The shifts observed in the R space may originate from the mixture of small Pt nanoparticles and single Pt atoms in different coordination environments. TEM images only provide tiny areas on a sample. I am not convinced that only single atom sites exist in these catalysts, considering the high temperatures (900 and 800 oC) used in the synthesis. How could the nucleation of Pt particles be prevented entirely when the samples were

annealed at 900 °C for 60 min? Further, the authors compared catalysts with different mass Pt loadings. Why can no Pt nanoparticles be formed in the 2.8 wt% sample while the 7 wt% sample has nanoparticles? Suppose you roughly estimate the specific surface area of carbon nanotubes. What would be the average distance between two Pt atoms if 2.8 wt% of Pt are all atomically dispersed on the carbon surface?

I would suggest that the better way to quantify the Pt-based active sites is to use CO chemisorption or electrochemical stripping (Current Opinion in Electrochemistry 2018, 9:198–206). Then, the authors could compare TOF of different active sites and draw conclusions related to their intrinsic activities.

Overall, I think the work may potentially provide useful information in the field. But the current experimental data and discussion are insufficient to support the claimed conclusions.

Reviewer #3 (Remarks to the Author):

The authors report here the oxygen reduction reaction pathway tuning over CNT supported Pt single atom catalysts coordinated with different metalloids, and their active sites density effect on products selectivity. The materials are well characterized by HAADF-STEM, EXAFS and XPS, which confirm the atomically dispersed feature of Pt central atoms and the different electronic structure as arisen from different coordination motifs. I love the logic and the comprehensive consideration of multiple perspectives, for both experimental setup and theoretical simulations of thermodynamics and kinetics investigation. Overall, the manuscript is well organized and the conclusion is well supported by the present results. It can be accepted at Nature Communications with a minor revision.

1. For Pt-N-C coordination, the annealing temperature effect on the ORR onset potential has been screened. How about Pt-S-C? Is there any room to further improve the H₂O₂ generation performance?
2. What's the sulfur and oxygen content within bare S-doped CNT? In Fig. S3, O signal in the Pt-S-CNT is significantly higher than other 2 samples, while earlier studies suggest that O-doped carbon can intrinsically convert O₂ into H₂O₂ at high activity, see Nat. Catal. 1, 156-162 (2018) and Nat. Catal. 1, 282-290 (2018) for reference. Relevant active motif assignment for either Pt central sites or O dopants should be properly addressed.
3. In Figure 4 of long-term electrolysis test, the leaching ratio of Pt should be quantitatively discussed.
4. In Experimental section, details regarding the sample preparation and testing condition for MicroCT should be supplemented.
5. XPS experimental data and the fitting results are not well consistent.
6. The authors used monolayer graphite model to describe the surface of CNT, ignoring the effect of curvature, which may affect the final conclusion.

Response to reviewers' comments:

We thank the reviewers and editor for constructive comments which have helped us to greatly improve our research and the quality of our manuscript. Below, we carefully address the points raised by reviewers one by one, with additional experimental inputs on statistic Pt sites distribution by ATR-IR and high-resolution XRD, site density and TOF analysis via electrochemical CO stripping, and quantitative Pt leaching, as well as theoretical simulations on the kinetic barrier comparison of $2e^-/4e^-$ ORR and potential substrate curvature effect.

Reviewer 1

“In this work, the authors prepared a series of Pt-X-C single atom catalysts and investigated the influence of local coordination environment on acidic oxygen reduction reaction pathway tuning. Isolated Pt-S motif is identified as the most selective one toward 2e pathway while 4e pathway is predominant on Pt-C motif. The experimental RRDE and bulk electrolysis researches combined with energetic analysis provide precise answer as to how the neighboring dopants influencing the reaction pathway, which is largely controversial in recent literatures. This present methodology and electrocatalytic results could make a great significance to the active motifs engineering in single atom catalysts as well as to shed light on the structure-activity relationship. I would suggest its publication at Nat. Commun. after addressing the comments below:”

--- We much appreciate the reviewer's overall positive evaluation to our work. Please see below our point-by-point response to each of the raised suggestion.

“1. For XPS spectra fitting, detailed parameters like the peak position, FWHM should be tabulated.”

--- Thanks for the suggestion, we have supplemented the detailed fitting parameters and tabulated in Supplementary Table 2, Page 28 of revised SI.

Supplementary Table 2. Detailed parameters for the core-level XPS spectra fitting.

Sample	XPS Peak		Position (eV)	FWHM (eV)	Area
200-Pt-C-CNT	Pt 4f _{7/2}		72.0	1.80	105.0
	Pt 4f _{5/2}		75.3	1.80	96.0
200-Pt-S-CNT	Pt 4f _{7/2}		72.5	2.00	168.5
	Pt 4f _{5/2}		75.8	2.00	126.4
	C-S-C	S 2p _{3/2}	164.2	0.98	547.3
		S 2p _{1/2}	165.4	0.98	273.6
	-SO _x	S 2p _{3/2}	165.9	0.98	58.3
		S 2p _{1/2}	167.0	0.98	29.2
200-Pt-N-CNT	Pt 4f _{7/2}		72.4	2.10	138.0
	Pt 4f _{5/2}		75.7	2.10	130.0
	N 1s	pyridine-N	398.5	1.28	158.3
		pyrrole-N	400.4	1.28	227.4
		graphitic-N	401.6	1.28	110.3
		oxidized-N	403.1	1.28	71.5
		Pt-N	399.2	0.98	105.0
400-Pt-N-CNT	Pt 4f _{7/2}		72.4	2.18	302.2
	Pt 4f _{5/2}		75.8	2.18	260.0
800-Pt-N-CNT	Pt 4f _{7/2}		72.4	2.25	782.8
	Pt 4f _{5/2}		75.8	2.25	672.0
2000-Pt-N-CNT	Pt ⁰	Pt 4f _{7/2}	71.5	0.87	139.6
		Pt 4f _{5/2}	74.8	0.87	115.1
	Pt ^{δ+}	Pt 4f _{7/2}	72.3	2.01	520.3
		Pt 4f _{5/2}	75.7	2.01	425.7

“2. In Supplementary Figure 12, the authors considered the kinetic barrier of Pt-S-C motif for either 4e or 2e pathway, for which the latter is significantly lower than the former, suggestive both thermodynamically and kinetically favorable O₂-to-H₂O₂ conversion. The authors are encouraged to supplement a similar kinetic analysis on Pt-C and Pt-N-C coordination structure.”

--- Revised as suggested. Using the same transition states, we further calculated the kinetic barrier of Pt-N₄ and Pt-C₄ motifs for both 4e⁻ and 2e⁻ pathway, please see below Fig. R1 for details.

Fig. R1. (a) Illustration of transition states by taking Pt-S₄ catalyzed ORR as an example, relevant intermediates and free energy diagrams for 2e⁻ (red) or 4e⁻ (black) ORR pathway on the Pt-S₄ (b), Pt-N₄ (c) and Pt-C₄ (d) motif. The kinetic barriers for *OOH to H₂O₂ or *O were computed by the Climbing Image Nudged Elastic Band (CI-NEB) approach.

For the kinetic analysis, we noted that Pt-S₄ exhibits the lowest kinetic energy barrier for 2e⁻ pathway (0.868 eV), followed by Pt-N₄ of 1.006 eV and Pt-C₄ of 2.024 eV.

For the thermodynamic analysis, we also plotted the Gibbs free energy change of OOH* to H₂O₂/O*, i.e., ΔG_{assoc} - ΔG_{dissoc}, as shown in Supplementary Figure 17:

Supplementary Figure 17. The relationship between the difference in Gibbs free energy change of OOH^* to $\text{H}_2\text{O}_2/\text{O}^*$ and the experimentally observed H_2O_2 selectivity.

In considering both the thermodynamics and kinetic barriers in DFT simulations, the theoretical tendency of $2e^-$ ORR pathway selectivity is $\text{Pt-S}_4 > \text{Pt-N}_4 > \text{Pt-C}_4$, in good agreement with our experimental observations. Besides, we have modified the above Supplementary Figure 17 on Page 18 of revised SI and Fig. 3 on Page 10 of revised Manuscript to fully address this reviewer's comment:

Fig. 3 DFT calculations of the ORR selectivity on different Pt-X-C moieties. (a) Examined configurations for Pt single atom coordinated in two-dimensional carbon matrix with different metalloids, *OOH adsorption is preferable on central Pt sites over neighboring S/N/C atoms. (b) Illustration of transition states taking Pt-S₄ as an example, (c-e) relevant free energy diagrams for 2e⁻ (red) or 4e⁻ (black) ORR pathway on the Pt-S₄ (c), Pt-N₄ (d) and Pt-C₄ (e) motif. Insert numbers represent the kinetic barriers for *OOH to H₂O₂ or *O, as computed by the Climbing Image Nudged Elastic Band (CI-NEB) approach.

“3. Regarding the H₂O₂ quantification from bulk electrolysis, what’s the detection limit for this KMnO₄ titration method, i.e., 1 ppm, 10 ppm?”

--- We appreciate the review’s careful note. To determine the detection limit of KMnO₄ titration, 1 μL of 30 wt.% H₂O₂ (GR, Sinopharm) was firstly diluted in 20 mL of Millipore water (ca. 16.6 ppm H₂O₂), then being added dropwise to a test tube containing 5 μL of 0.02 M KMnO₄ until the solution becomes colorless (Tube 1#, 2# and 3# in Supplementary Figure 24). We repeated it three

times, and the amount of H₂O₂ solution consumed was 505 μL, 510 μL and 507 μL, respectively, which was close to the theoretical value 510.5 μL that calculated from the reaction equation: $2\text{MnO}_4^- + 5\text{H}_2\text{O}_2 + 6\text{H}^+ \rightarrow 6\text{Mn}^{2+} + 5\text{O}_2 + 8\text{H}_2\text{O}$. The solution in the test tube of a control group (rightmost one) was still purple with 510 μL of water added, indicating that dilution does not make KMnO₄ colorless. To titrate the unknown H₂O₂ content in the bulk electrolysis solution, 1 mL of electrolyte was collected instead of the above standard 16.6-ppm H₂O₂ solution, therefore, the detection limit of this KMnO₄ titration can be calculated as $16.6 \times 510.5 / 1000 = 8.5$ ppm, which is lower than the first data point we measured in Figure 4e (ca. 71.7 ± 2.5 ppm from two independent measurements).

This result has been added as Supplementary Figure 24 on Page 25 of revised SI, and copied below for the reviewer's reference:

Supplementary Figure 24. H₂O₂ detection limit for KMnO₄ titration method. The left side is the KMnO₄ solution, and the right side is the solution after adding H₂O₂ or water.

“4. Open question, since O-doped carbon is widely reported to effectively convert O₂ into H₂O₂ in electrochemistry, how to differentiate the contribution of Pt active motif from O-doped CNT substrate?”

--- Thanks for the comment. We agree with the reviewer that O-doped carbon could potentially reduce O₂ into H₂O₂, as documented in earlier reports of *Nat. Catal.* **1**, 156-162 (2018), *Nat. Catal.* **1**, 156-162 (2018) and *Chem* **7**, 10.1016/j.chempr.2021.08.007 (2021). To differentiate the contribution of Pt coordinated motifs from the O-dopants in CNT substrate, we have compared the

oxygen content within Pt-S-CNT, bare S-CNT and O-CNT (adapted from *Nat. Catal.* **1**, 156-162 (2018)) and plotted their ORR performance to see if the H₂O₂ selectivity increases with increasing O content. Supplementary Figure 9a shows the XPS survey spectra with insert of tabulated surface species content. Though the O content in Pt-S-CNT is slightly higher than that in bare S-CNT, probably due to the secondary annealing of S-CNT with Pt precursor, these values are far less than the reported O-CNT of 8.2 at.%. Compared to the Pt-free counterparts, Pt-S-CNT shows the best electrocatalytic performance for O₂-to-H₂O₂ conversion, i.e., the onset potential positively shifts from 0.25 V to 0.46 V and the maximum H₂O₂ selectivity increases from 52.5% to 81.9% as compared to O_(8.2)-CNT, regardless the much higher O content in the latter. Therefore, we believe the Pt coordinated motif contributes mainly to the selective peroxide generation, especially at the low overpotential regime.

Additional XPS characterizations and relevant discussions have been supplemented as Supplementary Figure 9 on Page 10 of revised SI, and copied below for the reviewer's reference:

Supplementary Figure 9. (a) XPS survey spectra for Pt-S-CNT catalysts and S-CNT. The atomic content for each element, together with O-CNT as adapted from *Nat. Catal.* 1, 156-162 (2018), are tabulated as insert. (b) Linear sweep voltammetry of Pt-S-CNT, S-CNT and O-CNT recorded at 1600 rpm and a scan rate of 5 mV s^{-1} , together with the detected H_2O_2 currents on the ring electrode (upper panel) at a fixed potential of 1.2 V vs. RHE. The catalyst loading was fixed at 0.1 mg cm^{-2} . (c) Calculated H_2O_2 selectivity and electron transfer number during potential sweep. H_2O_2 selectivity and n were plotted from the onset potential that reached 0.1 mA cm^{-2} H_2O_2 partial current density. An earlier onset of $\sim 310 \text{ mV}$ plus an enhanced H_2O_2 selectivity is noted with the presence of Pt-S moiety, thus highlighting the promotion effect of Pt-S coordination to O_2 -to- H_2O_2 conversion.

Reviewer 2

“This manuscript describes the synthesis of Pt-(C, S, N)-CNT catalysts and their catalytic performance for 4e- pathway or 2e- pathway ORR. In terms of catalytic performance for ORR, the reported results are inferior to many recently published studies. The highest selectivity towards H₂O₂ is below 80%, much lower than many recent studies on Pt catalysts, oxidized carbon catalysts, and transition metal single-atom catalysts. Thus, the potential novelty of the current work is on the atomic structural tuning of single-atom sites and the understanding of their catalytic behaviors. Although different structures of M-N-C sites have been studied in several previous studies, the specific Pt-(C, S, N)-CNT comparison has not been reported before.”

--- We appreciate the reviewer’s overall evaluation on our work. Please see below our point-by-point response to each comment.

“The authors used high-temperature thermal treatment methods to prepare Pt-(C, S, N)-CNT catalysts. They also provided XAS, XPS, and TEM images to prove that only single atom sites were obtained. XAS provides the average coordination environment of bulk samples. The shifts observed in the R space may originate from the mixture of small Pt nanoparticles and single Pt atoms in different coordination environments. TEM images only provide tiny areas on a sample. I am not convinced that only single atom sites exist in these catalysts, considering the high temperatures (900 and 800 oC) used in the synthesis. How could the nucleation of Pt particles be prevented entirely when the samples were annealed at 900 oC for 60 min?”

--- Regarding the pyrolysis temperature for preparing atomic dispersed Pt catalysts over carbon substrates, they are actually adopted from previous reports, i.e., 800 °C for Pt-N-C moiety from *Nat. Commun.* **8**, 15938 (2017) and 900 °C for Pt-C moiety from *Angew. Chem. Int. Ed.* **58**, 1163-1167 (2019), which were cited as Ref. 37 and Ref. 38 in the Methods section, respectively. These literatures suggest that the strong interaction between Pt central atom and neighboring N or C dopant within carbon vacancies largely aid to anchor the Pt atomic sites, as reinforced by our STEM and XAFS characterizations. Moreover, we note that this high temperature annealing

method is widely adopted in preparing single atom catalysts, please see recent literatures on Rh₁/TiO₂ (650 °C under air atmosphere, *J. Am. Chem. Soc.* **143**, 16566–16579 (2021)) and FeN₄/C (800 °C under Ar atmosphere, *ACS Catal.* **11**, 13020–13027 (2021)) for example.

We agree with the reviewer that STEM images provide the dispersion information on localized area, while EXAFS spectra and fittings demonstrate the average coordination information of bulk sample. Though there're some uncertainties from experimental viewpoint, these two tools are the most widely recognized methodology for probing the metal distribution in single atom catalysts. To further probe if any Pt clusters and/or nanoparticles present in Pt-X-CNT, we have carried out additional high-resolution XRD (HR-XRD) and in situ electrochemical attenuated total reflection Infrared spectroscopy (ATR-IR) measurements with CO probe. Supplementary Figure 2 on Page 3 of revised SI plots the XRD patterns of 800 °C annealed Pt-N-CNT and 900 °C annealed Pt-C-CNT, together with bare CNT substrate as reference. To better illustrate if any Pt diffraction feature was embedded in the broad peak of carbon, we further performed the spectra subtraction using bare CNT as reference, and the results were shown as dotted lines in Supplementary Figure 2. Only graphitized C(020) peak was observed for all three samples at the absence of Pt(111) or Pt(200) diffraction peaks, thus suggestive there's no long range ordered Pt crystalline structure (nanoparticles).

Supplementary Figure 2. High resolution X-ray diffraction (XRD) patterns of Pt-N-CNT, Pt-C-CNT and CNT. The spectra were recorded on a Rigaku Mini Flex 600 spectrometer using a Cu K α radiation (40 kV, 15 mA) at a scan rate of 0.02° per step and a holding time of 4 s per step.

The insert histograms refer to standard Pt diffraction of PDF No. 00-004-0802 and C diffraction of PDF No. 00-054-0501.

Given the specific adsorption of CO on Pt and the very sensitive nature of $\nu(\text{CO})$ frequency to surface sites configuration, we have carried out in situ electrochemical ATR-IR measurements monitoring CO adsorption behavior on Pt-X-CNT and Pt film electrode. The time-course contour map of CO adsorption process is separately shown in Fig. R2a-R2c, while the spectra of CO_{ad} on Pt-X-CNT and bulk Pt electrode at 0.4 V in CO-saturated 0.1 M HClO_4 electrolyte are comparatively shown in Fig. R2d. On bulk Pt, both linear (CO_{L}) and bridge (CO_{B}) adsorption configuration are observed at 2082 and 1868 cm^{-1} , respectively, together with a linear bonded CO peak on Au underfilm (serving as conducting substrate and IR signal amplifier) centered at ~ 2018 cm^{-1} . While on Pt-X-CNT, only linear CO_{ad} peak is observed at the IR window from 1960 to 2040 cm^{-1} . For Pt-C-CNT and Pt-N-CNT, a pair of CO_{ad} peaks are noted, probably arisen from different coordination environment. The detailed spectroelectrochemical study as how to differentiate local coordination structure of M-X by comprehensive probe molecules and detection spectroscopies is beyond the scope of our present work, nevertheless, considering these STEM, EXAFS, HR-XRD and ATR-IR results, we believe highly dispersed Pt atoms over CNT substrate is the predominant dispersion feature in the as-prepared catalysts.

Fig. R2. Time-course ATR-IR contour maps of CO_{ad} on (a) Pt-S-CNT, (b) Pt-N-CNT and (c) Pt-C-CNT measured during 1800-s CO bubbling. (d) The comparison of CO_{ad} on Pt-S-CNT, Pt-N-CNT, Pt-C-CNT and bulk Pt film electrode at a fixed potential of 0.4 V in 0.1 M CO-saturated HClO₄ electrolyte. Please note that the CO_{ad} peak intensities observed on Pt-X-CNT and on Au-CO_L are quite different as mainly due to different surface enhancement arisen from Au underfilm, as well as the different extent of Au exposure during the catalyst ink coating.

On the other hand, we theoretically consider the ORR selectivity on Pt₁ versus Pt₆ cluster (J. Am. Chem. Soc. 133, 2541–2547 (2011)) anchored in S-doped carbon vacancy. These simulation results have been plotted as Supplementary Figure 18 on Page 19 of revised SI. The dissociative pathway is largely preferred on the Pt₆-S₄-C motif (ca. +2.83 eV energetic difference of $\Delta G_{\text{assoc}} - \Delta G_{\text{dissoc}}$) as compared to that on isolated Pt sites of Pt₁-S₄-C (ca. -0.71 eV). It is known that sub-

nanometric metal clusters are usually coordination unsaturated with M-M bond, the strong binding strength for oxygenated species on Pt_n clusters tends to break O-O bond and shifts the oxygen reduction toward the $4e^-$ pathway into H_2O product. Therefore, this simulation result in turn reinforces our assignment of highly dispersed Pt-X coordination motifs as active center for selective O_2 -to- H_2O_2 conversion.

Supplementary Figure 18. (a) Illustration of Pt_1-S_4-C and Pt_6-S_4-C motifs, together with the calculated free energy diagrams on (b) Pt_1-S_4 and (c) Pt_6-S_4 moieties for ORR process. The energetic difference between $2e^-$ and $4e^-$ ORR pathways ($\Delta G_{assoc} - \Delta G_{dissoc}$) is ca. -0.71 eV for the former and $+2.83$ eV for the latter, which means an energetic favorable O_2 -to- H_2O_2 conversion on Pt_1-S_4 moieties but a favorable O_2 -to- H_2O conversion on Pt_6-S_4 moieties.

We believe the above supplementary analysis from both experimental and theoretical efforts could reinforce our main conclusion that the Pt coordination structure rather than Pt nanoparticle dispersion is the main tuning knob that regulates ORR pathway.

“Further, the authors compared catalysts with different mass Pt loadings. Why can no Pt nanoparticles be formed in the 2.8 wt% sample while the 7 wt% sample has nanoparticles? Suppose you roughly estimate the specific surface area of carbon nanotubes. What would be the average distance between two Pt atoms if 2.8 wt% of Pt are all atomically dispersed on the carbon surface?”

--- Thanks for the comment. For the Pt-N-CNT sample with 2.8 wt.% Pt loading, the molar ratio of C:Pt is 564.4:1. If the carbon nanotube consists of 5 layers of graphitized carbon, the atomic ratio of C to Pt is about 113, which means that there are an average of 113 carbon atoms around a Pt central atom, ensuring the highly dispersed (or isolated) Pt feature at this loading. Please note that the present bright field TEM images in Fig. 5a are shown with 50-nm scale bar, thus a much larger view is present as compared to that in STEM images. Besides, we removed the legends of Pt single atoms and Pt nanoparticles in Fig. 5c to avoid any misunderstandings as suggested.

Fig. 5 Pt loading effect on H_2O_2 selectivity. (c) The maximum H_2O_2 selectivity as a function of Pt loading. The rightmost data point was measured on commercial TTK 30 wt.% Pt/C catalyst and plotted with actual Pt weight loading, while all other data on Pt-N-CNT were plotted using determined Pt content from XPS measurements.

“I would suggest that the better way to quantify the Pt-based active sites is to use CO chemisorption or electrochemical stripping (*Current Opinion in Electrochemistry* 2018, 9:198–

206). Then, the authors could compare TOF of different active sites and draw conclusions related to their intrinsic activities.”

--- We appreciate the review’s suggestion on quantifying active sites density and turnover frequency values. Electrochemical CO stripping measurements have been carried out as suggested. In brief, a Pt-X-CNT cast RDE working electrode began with bubbling CO (Air Liquid, >99.9% purity) in 0.1 M HClO₄ for 20 min at 0.25 V vs. RHE. Then, the dissolved CO was removed from the electrolyte by bubbling Ar for 60 min while maintaining the electrode potential at 0.1 V. Finally, we recorded the CO stripping voltammograms by scanning the potential from 0.25 to 1.30 V at 5 mV s⁻¹. A representative CO stripping voltammogram is plotted in Supplementary Figure 13, Page 14 of revised SI and adapted below for the reviewer’s reference.

Supplementary Figure 13. Representative cyclic voltammograms of electrochemical CO stripping on Pt-S-CNT, scan rate: 5 mV s⁻¹. The catalyst loading was fixed at 0.1 mg cm⁻². The CO_{des} charge integral interval is 0.7-1.3 V vs RHE, as marked in the blue shaded part.

According to Faraday's law, the amount of CO_{ad} can be calculated from its integrated charge. Besides, since CO_L is the predominant adsorption configuration on Pt surface as verified in the above ATR-IR spectra and early reports like *Chem. Soc. Rev.* **39**, 4643-4655 (2010), the electrochemical active Pt site density equals to the amount of CO_{ad}. The amount of Pt as derived from XPS and CO stripping results, as well as the theoretical loading calculated from raw ratio, are tabulated in Supplementary Table 3, Page 29 of revised SI and adapted below for the reviewer’s reference:

Supplementary Table 3. The amount of Pt measured by XPS and CO stripping.

sample	Theoretical loading (wt.%)	measured by XPS (wt.%)	measured by CO stripping (wt.%)
200-Pt-S-CNT	0.70	0.50	0.28
200-Pt-C-CNT	0.70	0.49	0.29
200-Pt-N-CNT_0.1mg cm ⁻²	0.70	0.66	0.29
200-Pt-N-CNT_0.2mg cm ⁻²	1.40	-	0.60
200-Pt-N-CNT_0.4mg cm ⁻²	2.80	-	1.26
400-Pt-N-CNT	1.40	0.96	0.36
800-Pt-N-CNT	2.80	2.68	0.47
2000-Pt-N-CNT	7.00	2.06	0.51

Along with the Pt active site quantification, the turnover frequencies for O₂-to-H₂O₂ and O₂-to-H₂O conversion as a function of applied potential are comparatively plotted in Supplementary Figure 14. The determined TOF_{SH₂O₂} on Pt-S-CNT and Pt-N-CNT are higher than TOF_{SH₂O}, while a reverse trend is observed on Pt-C-CNT. Besides, the TOF_{H₂O₂} on Pt-N-CNT is higher than that on Pt-S-CNT, which corresponds to the larger current density observed on Pt-N-CNT and therefore a higher intrinsic ORR activity on Pt-N as compared to Pt-S moiety. To better illustrate the selectivity trend, we further plot the ratio of TOF_{H₂O₂}/TOF_{H₂O} over different Pt coordination motifs. Based on this intrinsic TOF analysis, an increasing 2e⁻/4e⁻ ORR pathway selectivity is noted in the order of Pt-C-CNT < Pt-N-CNT < Pt-S-CNT, which is in good agreement with our previous RRDE results (Fig. 2b on Page 7 of revised MS) and theoretical simulations (Fig. 3 on Page 10 of revised MS).

Supplementary Figure 14. ORR performance of Pt-X-CNT catalysts cast RRDE in 0.1 M O₂-saturated HClO₄. (a) Linear sweep voltammograms of Pt-N-CNT (blue), Pt-S-CNT (red) and Pt-C-CNT (black) recorded at 1600 rpm and a scan rate of 5 mV s⁻¹, together with the detected H₂O₂ currents on the ring electrode (upper panel) at a fixed potential of 1.2 V vs. RHE. The catalyst loading was fixed at 0.1 mg cm⁻². (b) Calculated TOF and TOF_{H2O2}/TOF_{H2O} during LSV scan. All data were plotted from the onset potential that reached 0.1 mA cm⁻² H₂O₂ partial current density.

Fig. R3. (a) Calculated TOFs of 200-Pt-N-CNT with different catalyst loading (blue: 0.1 mg cm^{-2} , red: 0.2 mg cm^{-2} , black: 0.4 mg cm^{-2}) and (b) the detected H_2O_2 currents on the ring electrode at a fixed potential of 1.2 V vs. RHE , with a scan rate of 5 mV s^{-1} and at 1600 rpm . (c) Calculated TOFs of Pt-N-CNT with different electrochemical active Pt content (blue: $0.29 \text{ wt.}\%$, red: $0.36 \text{ wt.}\%$, black: $0.47 \text{ wt.}\%$, purple: $0.51 \text{ wt.}\%$ as derived from CO stripping measurements) but a fixed catalyst loading of 0.1 mg cm^{-2} , (d) the detected H_2O_2 currents on the ring electrode at a fixed potential of 1.2 V vs. RHE .

At last but not least, we calculated the TOFs under different Pt site densities using the same way. For 200-Pt-N-CNT, the TOF_{H₂O₂} decreases with increasing overall catalyst loading from 0.1 mg cm^{-2} to 0.4 mg cm^{-2} (Fig. R3), suggestive the negative effect of a longer diffusion path for H_2O_2

generation. Similarly, at a fixed catalyst loading of 0.1 mg cm^{-2} , the determined $\text{TOF}_{\text{H}_2\text{O}_2}$ decreases with increasing electrochemical active Pt site density from 0.29 wt.% to 0.51 wt.% as derived from CO stripping measurements, suggestive that the densely distributed Pt sites are beneficial for the total reduction of O_2 into H_2O rather than H_2O_2 generation.

These intrinsic TOFs analysis on the diffusion path effect and the Pt site density effect are actually in good agreement with our main conclusion as derived from electrochemical selectivity analysis. In this revision, we have cited relevant references of *JACS Au* **1**, 586–597 (2021), *Curr. Opin. Electrochem.* **9**, 198–206 (2018) as Refs. 52 and 53, compared TOFs of different Pt-X-CNT in Supplementary Figure 14, and supplemented relevant $\text{TOF}_{\text{H}_2\text{O}_2}$ plots into Fig. 5, Page 14 of revised MS, please see also the adapted figure below:

Fig. 5 Pt loading effect on H₂O₂ selectivity. (a) TEM images of 200-, 400-, 800- and 2000-Pt-N-CNT. Scale bars, 50 nm. Inserts are the zoom-in view of certain carbon nanotubes at a fixed scale bar of 10 nm. (b) Core-level XPS spectra for Pt 4f region. (c) The maximum H₂O₂ selectivity as a function of Pt loading. The rightmost data point was measured on commercial TKK 30 wt.% Pt/C catalyst and plotted with actual Pt weight loading, while all other data on Pt-N-CNT were plotted

using determined Pt content from XPS measurements. (d) Calculated TOFs of 200-Pt-N-CNT with different catalyst loading, and (e) calculated TOFs of Pt-N-CNT with different Pt site density but a fixed catalyst loading of 0.1 mg cm^{-2} .

“Overall, I think the work may potentially provide useful information in the field. But the current experiential data and discussion are insufficient to support the claimed conclusions.”

--- We thank again for the reviewer’s detailed comments that help us to polish our research and to greatly improve the quality of our manuscript. In summary of the above comments and responses, we have refined our main revisions below to further support our main conclusion on regulating oxygen reduction reaction pathway by the local coordination structure manipulation of highly dispersed Pt sites:

- 1) Regarding the dispersion feature of as-prepared Pt-X-CNT, our comprehensive experimental characterizations of atomic-resolved HAADF-STEM, synchrotron radiated EXAFS, *ex situ* HR-XRD and *in situ* electrochemical ATR-IR measurements with CO_{ad} probe have pointed out the high dispersion of isolated Pt sites over metallo-doped CNT substrates, which is also reinforced by the thermodynamic reaction pathway comparison between $\text{Pt}_1\text{-S}_4$ and $\text{Pt}_6\text{-S}_4$ representative moieties.
- 2) Regarding the quantification of Pt active site density and turnover frequency, electrochemical CO stripping measurements and relevant intrinsic TOF calculations have been carried out as suggested. It turns out that Pt-S-C delivers the highest $2e^-$ ORR selectivity while Pt-N-C delivers the highest intrinsic ORR activity. Further analysis on the Pt loading effect indicates that a short diffusion path as well as a low Pt site density could aid to the electrochemical synthesis of H_2O_2 , which reinforces our previous conclusion on the active site assignment and Pt site density effect.

We believe the present work combining comprehensive catalyst characterization and detailed structure-activity relationship investigation could stimulate the research community and shed light on the design of more efficient electro-catalysts.

Reviewer 3

“The authors report here the oxygen reduction reaction pathway tuning over CNT supported Pt single atom catalysts coordinated with different metalloids, and their active sites density effect on products selectivity. The materials are well characterized by HAADF-STEM, EXAFS and XPS, which confirm the atomically dispersed feature of Pt central atoms and the different electronic structure as arisen from different coordination motifs. I love the logic and the comprehensive consideration of multiple perspectives, for both experimental setup and theoretical simulations of thermodynamics and kinetics investigation. Overall, the manuscript is well organized and the conclusion is well supported by the present results. It can be accepted at Nature Communications with a minor revision.”

--- We much appreciate the reviewer’s overall positive comment on our work. Please see below the point-by-point response to the reviewer’s suggestion.

“1. For Pt-N-C coordination, the annealing temperature effect on the ORR onset potential has been screened. How about Pt-S-C? Is there any room to further improve the H₂O₂ generation performance?”

--- In this revision, we have screened the annealing temperature effect on the ORR pathway tuning for Pt-S-CNT as suggested. As shown in Supplementary Figure 10, as the annealing temperature increases from 150 °C to 350 °C, the current density increases, and the onset potential shifts positively. Compared to the pristine Pt-S-CNT prepared at 250 °C, the maximum H₂O₂ selectivity now increases from 81.9% to 88.9% with optimized annealing temperature. Nevertheless, the H₂O₂ generation performance decreases with further temperature increasing up to 450 °C, possibly due to the escaping of S content at elevated temperature.

Supplementary Figure 10. ORR performance of Pt-S-CNT catalysts prepared at different annealing temperature. (a) Linear sweep voltammetry of Pt-S-CNT recorded at 1600 rpm and a scan rate of 5 mV s^{-1} , together with the detected H_2O_2 currents on the ring electrode (upper panel) at a fixed potential of 1.2 V vs. RHE. The catalyst loading was fixed at 0.1 mg cm^{-2} . (b) The onset potential of Pt-S-CNT catalysts annealed at different temperature and (c) potential-dependence H_2O_2 selectivity. The onset potential was defined as that delivers 0.1 mA cm^{-2} H_2O_2 partial current density.

This supplementary figure has been added into Page 11 of revised SI with relevant data point and discussions have been updated into the performance map of Fig. 2c, Page 7-8 of revised MS.

“2. What’s the sulfur and oxygen content within bare S-doped CNT? In Fig. S3, O signal in the Pt-S-CNT is significantly higher than other 2 samples, while earlier studies suggest that O-doped

carbon can intrinsically convert O₂ into H₂O₂ at high activity, see Nat. Catal. 1, 156-162 (2018) and Nat. Catal. 1, 282-290 (2018) for reference. Relevant active motif assignment for either Pt central sites or O dopants should be properly addressed.”

--- Thanks for pointing out the potential issue of O content effect in determining H₂O₂ selectivity. We agree with the reviewer that O-doped carbon could potentially reduce O₂ into H₂O₂, as documented in those earlier reports. To better differentiate the contribution of Pt coordinated motifs from the O-dopants in CNT substrate, we have compared the oxygen content within Pt-S-CNT, bare S-CNT and O-CNT (adapted from *Nat. Catal.* **1**, 156-162 (2018)) and plotted their ORR performance to see if the H₂O₂ selectivity increases with increasing O content. Supplementary Figure 9a shows the XPS survey spectra with insert of tabulated surface species content. Though the O content in Pt-S-CNT is slightly higher than that in bare S-CNT, probably due to the secondary annealing of S-CNT with Pt precursor, these values are far less than the reported O-CNT of 8.2 at.%. Compared to the Pt-free counterparts, Pt-S-CNT shows the best electrocatalytic performance for O₂-to-H₂O₂ conversion, i.e., the onset potential positively shifts from 0.25 V to 0.46 V and the maximum H₂O₂ selectivity increases from 52.5% to 81.9% as compared to O_(8.2)-CNT, regardless the much higher O content in the latter. Therefore, we believe the Pt coordinated motif contributes mainly to the selective peroxide generation, especially at the low overpotential regime.

Additional XPS characterizations and relevant discussions have been supplemented as Supplementary Figure 9 on Page 10 of revised SI, and copied below for the reviewer's reference:

Supplementary Figure 9. (a) XPS survey spectra for Pt-S-CNT catalysts and S-CNT. The atomic content for each element, together with O-CNT as adapted from *Nat. Catal.* **1**, 156-162 (2018), are tabulated as insert. **(b)** Linear sweep voltammetry of Pt-S-CNT, S-CNT and O-CNT recorded at 1600 rpm and a scan rate of 5 mV s^{-1} , together with the detected H_2O_2 currents on the ring electrode (upper panel) at a fixed potential of 1.2 V vs. RHE. The catalyst loading was fixed at 0.1 mg cm^{-2} . **(c)** Calculated H_2O_2 selectivity and electron transfer number (n) during potential sweep. H_2O_2 selectivity and n were plotted from the onset potential that reached 0.1 mA cm^{-2} H_2O_2 partial current density. An earlier onset of $\sim 310 \text{ mV}$ plus an enhanced H_2O_2 selectivity is noted with the presence of Pt-S moiety, thus highlighting the promotion effect of Pt-S coordination to O_2 -to- H_2O_2 conversion.

“3. In Figure 4 of long-term electrolysis test, the leaching ratio of Pt should be quantitatively discussed.”

--- Revised as suggested, we have performed additional ICP-MS measurement on a Thermo Scientific iCAP-Q Inductively Coupled Plasma Mass Spectrometry to quantify the potential Pt leaching ratio during electrolysis with a detection limit of 0.001 ppb. The electrolyte was picked up from repeated long-term ORR stability test over Pt-S-CNT sample (Fig. 4d in Manuscript), for which 0.5 mg catalyst (or ca. 3.5 μg of Pt) and 25 mL of 0.1 M HClO_4 solution was used. If all the Pt was leached out from catalyst, the Pt concentration in electrolyte should be ~ 140 ppb.

Supplementary Figure 21 shows the time-evolved Pt content in electrolyte, after 10 hours continuous electrolysis, only 0.40 ppb Pt is leached out, corresponding to a molar ratio of 0.29 at.%. We believe the intrinsic stable coordination configuration, as well as the operation potential (i.e. < 0.1 V vs. RHE thus below the redox potential of $\text{Pt}^{2+}/\text{Pt}^0$), contribute mainly to this observed structural stability.

Supplementary Figure 21. Time-evolved Pt leaching during the long-term electrolysis as shown in Fig. 4d. After 10 hours' continuous electrolysis, 0.40 ppb Pt was leached from Pt-S-CNT catalyst, corresponding to 0.29 at% molar ratio.

Relevant discussions and supplementary figures have been added onto Page 12 of revised MS and Page 22 of revised SI.

“4. In Experimental section, details regarding the sample preparation and testing condition for MicroCT should be supplemented.”

--- Revised as suggested. Relevant experimental details have been supplemented into Material Synthesis and Characterization section, Page 17-18 of revised MS, and copied below for the reviewer’s reference:

“The catalysts morphology was characterized by a FEI Sirion 200 field-emission scanning electron microscopy (SEM), using an electron beam energy of 5 kV and a spot size of 3.0 nm with magnification ranging from 5 to 80 k. MicroCT was performed on a Zeiss Xradia 520 Versa X-ray microscopy. The distances of the sample to the X-ray source (12.53 mm) or to the X-ray detector (72.09 mm) result in a voxel (volume pixel) size of 1 μm . The field of view (FOV) was approximately 1012.3 μm \times 1012.3 μm , and the reconstruction of MicroCT data was conducted using the TXM Reconstructor software (Xradia).”

“5. XPS experimental data and the fitting results are not well consistent.”

--- Regarding to the XPS spectra fitting, we strictly followed the criteria, relevant fitting details have now been supplemented in Supplementary Table S2, Page 28 of revised SI and also copied below for the reviewer’s reference:

Supplementary Table 2. Detailed parameters for the core-level XPS spectra fitting.

Sample	XPS Peak	Position (eV)	FWHM (eV)	Area	
200-Pt-C-CNT	Pt 4f _{7/2}	72.0	1.80	105.0	
	Pt 4f _{5/2}	75.3	1.80	96.0	
200-Pt-S-CNT	Pt 4f _{7/2}	72.5	2.00	168.5	
	Pt 4f _{5/2}	75.8	2.00	126.4	
	C-S-C	S 2p _{3/2}	164.2	0.98	547.3
		S 2p _{1/2}	165.4	0.98	273.6
	-SO _x	S 2p _{3/2}	165.9	0.98	58.3
		S 2p _{1/2}	167.0	0.98	29.2

200-Pt-N-CNT	Pt 4f _{7/2}		72.4	2.10	138.0
	Pt 4f _{5/2}		75.7	2.10	130.0
	N 1s	pyridine-N	398.5	1.28	158.3
		pyrrole-N	400.4	1.28	227.4
		graphitic-N	401.6	1.28	110.3
		oxidized-N	403.1	1.28	71.5
Pt-N		399.2	0.98	105.0	
400-Pt-N-CNT	Pt 4f _{7/2}		72.4	2.18	302.2
	Pt 4f _{5/2}		75.8	2.18	260.0
800-Pt-N-CNT	Pt 4f _{7/2}		72.4	2.25	782.8
	Pt 4f _{5/2}		75.8	2.25	672.0
2000-Pt-N-CNT	Pt ⁰	Pt 4f _{7/2}	71.5	0.87	139.6
		Pt 4f _{5/2}	74.8	0.87	115.1
	Pt ^{δ+}	Pt 4f _{7/2}	72.3	2.01	520.3
		Pt 4f _{5/2}	75.7	2.01	425.7

Given the strict requisites of $\Delta B.E.$, FWHM and relevant peak ratio between each component, we tried our best to perform a rationalized and accurate data interpretation without overfitting.

“6. The authors used monolayer graphite model to describe the surface of CNT, ignoring the effect of curvature, which may affect the final conclusion.”

--- Thanks for directing us to this theoretical model issue. The curved carbon nanotube model contains 3~5 times more atoms than that in monolayer graphite model, which consumes an excess of computing resource. As an alternative, the simplified 2D monolayer model is widely deployed in literatures to theoretically mimic the active sites structure in metal or metalloid doped CNTs for electrolysis. Please see the adapted pictures below for example:

Fig. R4 Representative models of monolayer graphite to mimic CNT surface in electrocatalysis studies. Those models are adapted from (a) Lu, Z. et al. High-efficiency oxygen reduction to hydrogen peroxide catalyzed by oxidized carbon materials. *Nat. Catal.* **1**, 156-162 (2018), (b) Wang, J. et al. Synergistic effect of well-defined dual sites boosting the oxygen reduction reaction. *Energy Environ. Sci.* **11**, 3375-3379 (2018), (c) Kim, J. H. et al. A general strategy to atomically dispersed precious metal catalysts for unravelling their catalytic trends for oxygen reduction reaction. *ACS Nano* **14**, 1990-2001 (2020).

To further evaluate the potential effect of CNT curvature on ORR selectivity, we have performed additional calculations on the thermodynamical free energies of O₂-to-H₂O₂ conversion using Pt-S₄ motif anchored on either monolayer graphite or curved carbon nanotube substrates as a representative. Their configurations and free energy diagrams are comparatively plotted in Supplementary Figure 25, Page 26 of revised SI. Nearly identical adsorption energies on *OOH intermediate is observed, thus suggestive the minor curvature effect on H₂O₂ generation.

*Supplementary Figure 25. Curvature effect evaluation on O₂-to-H₂O₂ conversion. (a) Illustration of Pt-S₄ motif anchored on either monolayer graphite or (b) curved carbon nanotube substrate. (c) Calculated free energy diagrams on Pt-S-Graphite and Pt-S-CNT moieties. Nearly identical *OOH adsorption strength is revealed, suggestive a minor curvature effect on H₂O₂ generation.*

REVIEWERS' COMMENTS

Reviewer #1 (Remarks to the Author):

From the revised manuscript and the response to the comments, I found that the authors ahve dealt with all my concerns very well by supplementing lots of new experimental results and discussions, now it can be accepted for publication.

Reviewer #2 (Remarks to the Author):

The revision has addressed my early comments adequately. I support its publication as it is.

Reviewer #3 (Remarks to the Author):

All the questions proposed by the reviewers are well answered by the authors. The revised manuscript is acceptable to be published on Nature communications.

Response to reviewers' comments:

We thank the reviewers and editor for constructive comments which have helped us to greatly improve our research and the quality of our manuscript.

Reviewer 1

“From the revised manuscript and the response to the comments, I found that the authors have dealt with all my concerns very well by supplementing lots of new experimental results and discussions, now it can be accepted for publication.”

--- We much appreciate the reviewer's recognition of our work.

Reviewer 2

“The revision has addressed my early comments adequately. I support its publication as it is.”

--- We much appreciate the reviewer's recognition of our work.

Reviewer 3

“All the questions proposed by the reviewers are well answered by the authors. The revised manuscript is acceptable to be published on Nature communications.”

--- We much appreciate the reviewer's recognition of our work.